# Spatial transcriptomics reveals metabolic changes underly age-dependent declines in digit regeneration

**Robert J Tower**[1]*[†], **Emily Busse**[2], **Josue Jaramillo**[2], **Michelle Lacey**[3], **Kevin Hoffseth**[4], **Anyonya R Guntur**[5], **Jennifer Simkin**[6], **Mimi C Sammarco**[2]*

[1]Department of Orthopaedics, Johns Hopkins University, Baltimore, United States; [2]Department of Surgery, Tulane School of Medicine, New Orleans, United States; [3]Department of Mathematics, Tulane University, New Orleans, United States; [4]Department of Biological & Agricultural Engineering, Louisiana State University, Baton Rouge, United States; [5]Center for Molecular Medicine, Maine Medical Center Research Institute, Scarborough, United States; [6]Department of Orthopaedic Surgery, Louisiana State University Health Sciences Center, New Orleans, United States

**\*For correspondence:**
robert.tower@utsouthwestern.edu (RJT);
msammarc@tulane.edu (MCS)

**Present address:** [†]Department of Surgery, University of Texas Southwestern Medical Center, Dallas, United States

**Competing interest:** The authors declare that no competing interests exist.

**Abstract** De novo limb regeneration after amputation is restricted in mammals to the distal digit tip. Central to this regenerative process is the blastema, a heterogeneous population of lineage-restricted, dedifferentiated cells that ultimately orchestrates regeneration of the amputated bone and surrounding soft tissue. To investigate skeletal regeneration, we made use of spatial transcriptomics to characterize the transcriptional profile specifically within the blastema. Using this technique, we generated a gene signature with high specificity for the blastema in both our spatial data, as well as other previously published single-cell RNA-sequencing transcriptomic studies. To elucidate potential mechanisms distinguishing regenerative from non-regenerative healing, we applied spatial transcriptomics to an aging model. Consistent with other forms of repair, our digit amputation mouse model showed a significant impairment in regeneration in aged mice. Contrasting young and aged mice, spatial analysis revealed a metabolic shift in aged blastema associated with an increased bioenergetic requirement. This enhanced metabolic turnover was associated with increased hypoxia and angiogenic signaling, leading to excessive vascularization and altered regenerated bone architecture in aged mice. Administration of the metabolite oxaloacetate decreased the oxygen consumption rate of the aged blastema and increased WNT signaling, leading to enhanced in vivo bone regeneration. Thus, targeting cell metabolism may be a promising strategy to mitigate aging-induced declines in tissue regeneration.

## Editor's evaluation

This paper will be of interest to a broad range of scientists in the regeneration field as it builds on and complements several recent mouse digit tip regeneration single-cell RNAseq data sets. This study applies the emerging field of spatial transcriptomics to overlay gene expression information on the spatial arrangement of regenerating cells over time. The authors use their data to address several important questions related to regeneration such as "can we define a molecular signature for regenerating cells?" and "why does regenerative ability decline with age?

## Introduction

The ability to regenerate lost or damaged limb structures de novo, where new growth replaces both the amputated bone and surrounding soft tissue, is a highly desirable biological process that varies widely in vertebrates. While wound repair is highly conserved in all organisms, the ability to regenerate complex limb structures is limited and, in mammals, restricted to the distal one-third of the third phalangeal element (P3). This regenerative response is specific to the level of amputation: amputation of the distal one-third of the terminal phalangeal element (P3) can regenerate, whereas an amputation through the middle phalangeal element (P2) forms a hypertrophic callus of the bone. This regenerative response has been well documented in rodents, monkeys, and humans (*Bryant et al., 2002*; *Brockes and Kumar, 2005*; *Han et al., 2008*; *Fernando et al., 2011*; *Simkin et al., 2013*). Central to this process is the dedifferentiated tissue structure called the blastema that will ultimately give rise to the regenerated structure. Regeneration follows a defined sequence, including (i) initial injury and inflammatory response (degradation and epidermal closure days 7–10), (ii) anabolic cell dedifferentiation, recruitment, and proliferation during formation of the critical blastema (day 10), and (iii) differentiation (day 14 and later) in which the bone is regenerated via direct intramembranous ossification (*Han et al., 2008*; *Fernando et al., 2011*; *Simkin et al., 2013*). These distinct and independent phases of regeneration lead to bone produced via direct ossification and provide a controlled model to study bone regeneration within the context of aging.

Energetic demands are affected by cell type and function, local environment, and a cell's progression along the differentiation pathway. As a result, proliferating skeletal progenitors and differentiated osteoblasts exhibit different levels and types of cell metabolism (*van Gastel and Carmeliet, 2021*). Proliferating progenitors require ATP for energy and carbon for biomass (*Lunt and Vander Heiden, 2011*) and are highly glycolytic (*Salazar-Noratto et al., 2020*). This glycolytic dependency increases in differentiated osteoblasts due to the high metabolic demand of matrix and amino acids synthesis (*van Gastel and Carmeliet, 2021*; *Capulli et al., 2014*; *Bonewald, 2011*; *Lee et al., 2020*). The aging process disrupts cell metabolism by decreasing $NAD^+$ levels (*Neri and Borzì, 2020*; *Song et al., 2019*) and mitochondrial function and content (*Yu et al., 2018*; *Bellantuono et al., 2009*), leading to a loss of osteogenic potential. Prior research such as Seahorse analysis has allowed us to analyze mitochondrial respiration and glycolysis in live cells in vitro as it relates to osteoblast function. However, a better understanding of how cell metabolism regulates gene expression during aging, and how these changes can be modulated to benefit potential regenerative strategies in the future, is needed.

Digit regeneration via direct ossification taps into an endogenous bone regrowth process that, once dissected, has the potential to be impactful in many bone disease and injury models. Significant effort has been placed on dissecting out the essential regenerative signaling pathways for their application in non-regenerative models and tissue engineering-based approaches. Up until recently, this has proven technically challenging owing to an inability to accurately interrogate the transcriptional profile of the blastema. To this end, recent papers have made use of single-cell RNA sequencing (scRNAseq) to generate a blastema-specific gene signature (*Storer et al., 2020*; *Johnson et al., 2020*). While these published works were able to provide incredible insight into the cellular heterogeneity of the blastema, scRNAseq is unable to provide spatial context to these identified transcriptional profiles. Identifying the spatial localization of the signals that drive limb regeneration is a critical piece in discerning the mechanisms that drive limb patterning and the timing of differentiation during this multi-tissue process (*Bassat and Tanaka, 2021*; *Storer and Miller, 2020*; *Johnson and Lehoczky, 2022*).

Here, for the first time, we make use of spatial transcriptomics to investigate the spatially defined transcriptional profile of the regenerating blastema. Using this approach, we generated a gene signature for the blastema with high specificity and fidelity across published transcriptomic data sets (*Storer et al., 2020*; *Johnson et al., 2020*). To further characterize the blastema, we utilized aging as a model of impaired regeneration. Comparative spatial analysis revealed multiple age-dependent changes in pathway activation transcripts, including changes in proliferation and cell metabolism. Hypothesizing that age-dependent impaired regeneration was a function of dysregulated cell metabolism, we sought to modulate this response using exogenous metabolites. Administration of the metabolite oxaloacetate (OAA) reversed bone thickness and volume deficits observed in aged mice. Spatial transcriptomic analysis suggests that this rescue may be mediated through increased expression of bioenergetic transcripts and WNT pathway activation. These findings suggest that cell metabolism

underpins many aspects of tissue regeneration and underscores the potential of metabolite intermediates to enhance and modify regenerative capacity.

## Results
### Spatial transcriptomics defines the regenerating blastema

Traditionally, identifying molecular markers specific to the blastema has been difficult because of the heterogeneity of cell types present in the blastema, as well as the shared expression of previously identified markers (i.e. developmental genes such as *Msx1*, stem-like genes such as *Cxcl12*) with other dermal cells (*Storer and Miller, 2020*). Instead, the blastema is defined histologically within the new growing tissue area (*Storer and Miller, 2020*). In light of this, we amputated the distal one-third of the third phalangeal element (P3), allowed formation of the regenerative blastema (D10), then made use of spatial transcriptomics to provide a whole transcriptome characterization of the blastema as a complete structure within the context of its surrounding tissue in situ. This technique makes use of gene expression slides coated in poly-T primers and encoded with a unique spatial barcode. Fresh frozen sections are placed on a tissue optimization slide and subjected to varying times of enzymatic permeabilization to release cellular mRNA. Captured mRNA is visualized through incorporation of fluorescent nucleotides during on-slide cDNA synthesis (*Figure 1—figure supplement 1A*). To assess gene expression, a similar protocol is followed using pre-determined optimal digestion times. Following library preparation and sequencing, reads are aligned, and the unique spatial barcode is used to determine the spatial location of origin for each transcript, registered to the original hematoxylin and eosin (H&E) stained image (*Figure 1—figure supplement 1B*). This technique yielded ~4500 spatial 'spots' containing ~1900 unique genes (nFeature) and ~5500 unique transcripts (nCount) per spot (*Figure 1—figure supplement 1C*) at a spatial resolution of 55 µm, corresponding to ~5–10 cells per spatial spot, depending on the relative location within the blastema. Spatially restricted gene expression and image registration was confirmed by using positive marker genes for the epithelium (*Krt14*) and bone (*Bglap2*), by expression of previously proposed blastema markers *Pdgfra* (*Storer et al., 2020*) and *Mest* (*Johnson et al., 2020*; *Figure 1A*). Conversely, although hematopoietic cells infiltrate the digit at relatively high numbers after injury (*Simkin et al., 2017*), these cells are present in a more diffuse pattern, surrounded by other cell types. This resulted in minimal detection of the hematopoietic cell marker CD45 (encoded by *Ptprc*) (*Figure 1A*). Owing to the registration of transcriptomic data to histological images, spatial transcriptomics allows us to spatially define the area of the blastema relative to the rest of the digit. This allowed for the manual segmentation and identification of the blastema, boundary (spots overlaying both the blastema and surrounding connective tissue), and the remaining digit structure using morphological parameters (*Figure 1B*). Differential gene expression was used to determine those genes showing enriched expression within the spatially defined blastema, boundary, and remaining digit regions (*Supplementary files 1-3*). These results identified 829 differentially expressed genes (DEGs) enriched in the blastema, 146 of which were also enriched within the 337 boundary spot DEGs, and no overlap with the 214 DEGs identifying the remaining digit (*Figure 1C*). Interestingly, pathway analysis on blastema DEGs showed enrichment in gene ontology terms linked to cellular bioenergetics, in addition to major signaling pathways such as transforming growth factor beta (TGFβ), PI3K-Akt, and MAPK (*Figure 1D*).

From our blastema DEG list, we generated a 100 gene 'blastema signature' using genes enriched in the blastema (*Supplementary file 4*, listed in descending order of blastema specificity). Spatial transcriptomics blastema signature genes were selected based first on the regional specificity for the blastema, then on log fold change in gene expression between the blastema and surrounding tissue, and finally for adjusted statistical significance (see Materials and methods for details). Module scoring, reflecting the average expression of a selected gene list relative to random gene list controls, and spatial localization of our blastema signature showed a high degree of specificity for the regenerating blastema and minimal labeling of the remaining digit tissue (*Figure 1E*). Recently, two scRNAseq studies (*Storer et al., 2020*; *Johnson et al., 2020*) have proposed genetic signatures of the blastema from micro-dissected tissue. To validate our spatially derived blastema signature, we cross-compared our blastema signature with (*Storer et al., 2020*) and (*Johnson et al., 2020*) scRNAseq-defined blastema signatures (*Supplementary file 4*). Briefly, we used the same selection criteria that was applied to our spatial transcriptomics blastema signature, which excluded genes from cluster 8 enriching for

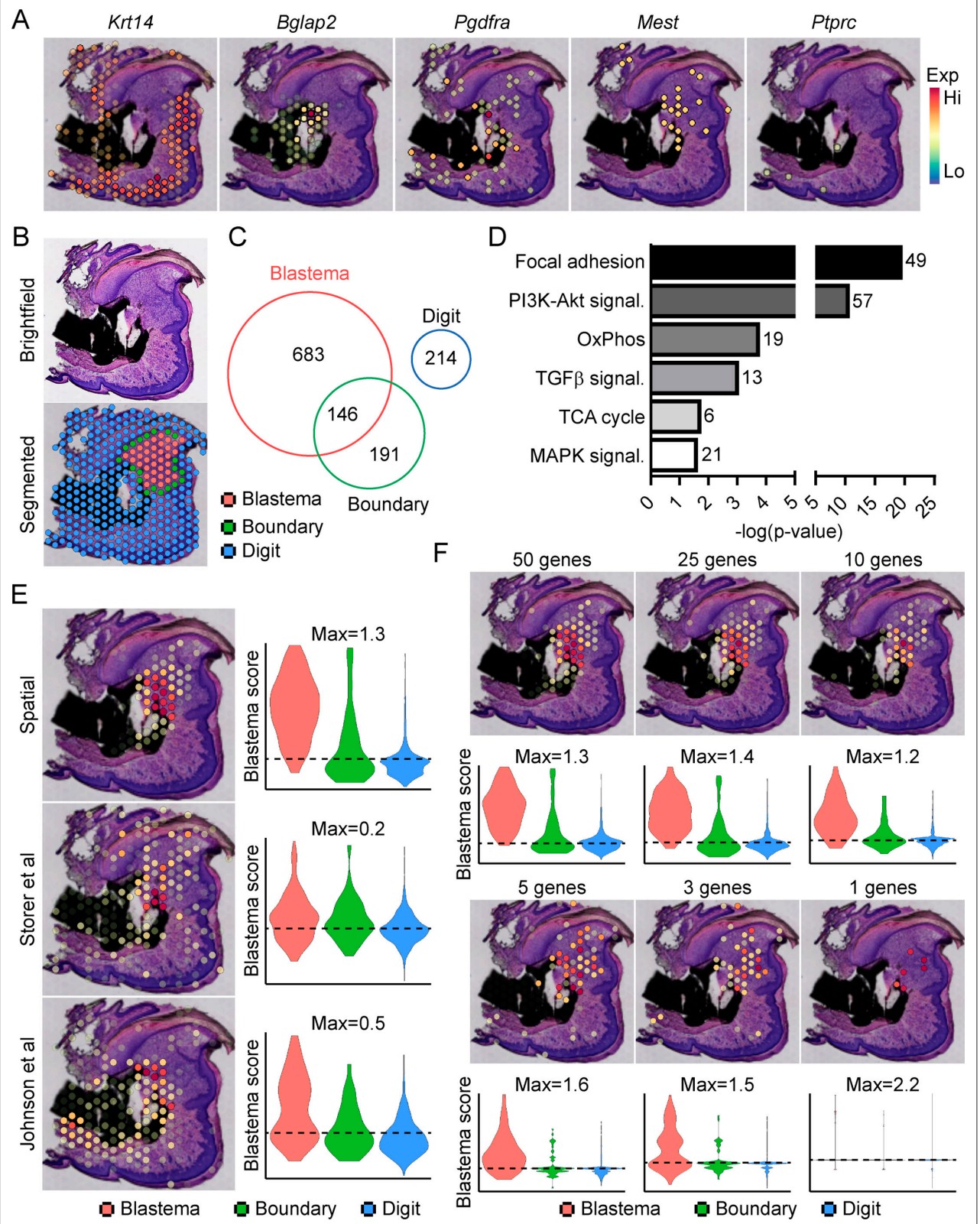

**Figure 1.** Transcriptional profiling of the blastema using spatial transcriptomics. (**A**) Representative spatial gene expression of epithelial (*Krt14*), bone (*Bglap2*), blastema (*Pdgfra*, *Mest*), and hematopoietic cell (CD45, encoded by *Ptprc*) transcripts. (**B**) Manual segmentation of spatial data into blastema (red), boundary (green), and remaining digit spots (blue) based on histological guidance. (**C**) Differentially expressed genes (DEGs) showing enriched expression in the blastema, boundary spots, or remaining digit. (**D**) Pathway analysis of genes showing enriched expression in the blastema. Number

*Figure 1 continued*

at end of bar denotes the number of significantly regulated genes assigned to each indicated KEGG term. (**E**) Module scoring of top 100 blastema genes derived from spatial transcriptomics or single-cell RNA sequencing (scRNAseq) (*Storer et al., 2020*; *Johnson et al., 2020*). (**F**) Blastema module scoring based on decreasing numbers of genes as indicated. Dotted lines denote a blastema modular score of 0 with higher values indicative of enrichment for the blastema fingerprint.

The online version of this article includes the following figure supplement(s) for figure 1:

**Figure supplement 1.** Overview of spatial transcriptomics approach.

**Figure supplement 2.** Exclusion of cell cycling cells identified in fibroblast cluster 8 from *Johnson et al., 2020*, improves labeling proficiency of the spatial blastema footprint.

**Figure supplement 3.** Comparison of blastema gene signatures in single-cell RNA sequencing (scRNAseq) data sets.

proliferative genes (*Figure 1—figure supplement 2*), to identify a blastema gene signature from each original data set. Genes from Storer et al. were selected from category 1 identified in Table S2 of the original publication (*Storer et al., 2020*), and genes from *Johnson et al., 2020* were selected from fibroblast-enriched genes identified in Table S1A. All blastema signatures showed preferential enrichment, to varying extents, within our blastema region relative to other regions of the digit, further validating this spatial transcriptomics technique (*Figure 1E*).

To determine the widespread feasibility of our spatial blastema signature to label blastema fibroblasts, we next conducted the reciprocal analysis, applying our blastema gene signature derived from spatial back to the two original sets of scRNAseq data (*Figure 1—figure supplement 3*). ScRNAseq data sets derived from Storer et al. (D10/14) (*Figure 1—figure supplement 3A*) and Johnson et al. (D11) (*Figure 1—figure supplement 3B*) were reanalyzed to generate UMAP similar to those presented within the original publications. Clusters circled in red denote fibroblast-like clusters used to define the blastema by the original authors. Module scoring confirmed the spatial transcriptomics-derived blastema gene signature had a high specificity for the blastema in both scRNAseq data sets, distinguishing the blastema fibroblasts from other cell types, and specifically excluded more mature osteogenic clusters (cluster 6 from Storer et al. data and cluster 5 from Johnson et al. data). Following validation of our blastema gene signature, we next applied reductive iterations to our spatial blastema gene list in order to identify a threshold of signature genes required to successfully label the blastema (*Figure 1F*). These results demonstrate that, although no single gene was able to show efficient labeling of the entire blastema, specific labeling of the blastema could be achieved using a reduced gene list of ~5 genes (*Edil3*, *Mmp14*, *Chrdl1*, *Ptx3*, and *Bmp5*). Taken together, these data demonstrate that spatial transcriptomics can successfully delineate gene expression in the blastema, and illustrate the widespread applicability of spatially derived data to other published transcriptomic data sets.

## Aged mice show a reduced regenerative potential associated with altered blastema bioenergetics

Having established the ability of spatial to successfully delineate the transcriptional profile of the blastema, we next sought to apply this technique in a model of impaired regeneration to understand potential regulatory mechanisms essential for successful digit regeneration. Owing to the well-established, age-dependent impediments in bone and wound repair (*Gurtner et al., 2008*; *Coffman et al., 2016*), we investigated the effects of aging on digit regeneration. We amputated the distal one-third of P3 in young (6 months of age) and aged (18 months of age) mice and monitored over time using micro-computed tomography (micro-CT) and histological assessments (*Figure 2A and B*). These data revealed that, while aged mice showed no difference in the timing of the histolytic event (D7), a marked delay in the transition from blastema to anabolic bone formation was observed. While new bone was observed at D14 in young mice, new bone wasn't observed in aged mice until D21. This delay in mineralization was associated with a reduction in trabecular bone volume and thickness, and a concordant increase in trabecular spacing (*Figure 2C*). These data confirmed that, similar to previous results in wound and bone regeneration (*Seifert and Voss, 2013*), aging results in a significant reduction in digit regeneration following amputation.

To determine the potential mechanisms underlying this impaired regeneration in aged mice, we conducted spatial transcriptomics on the aged blastema as described above. We used comparative

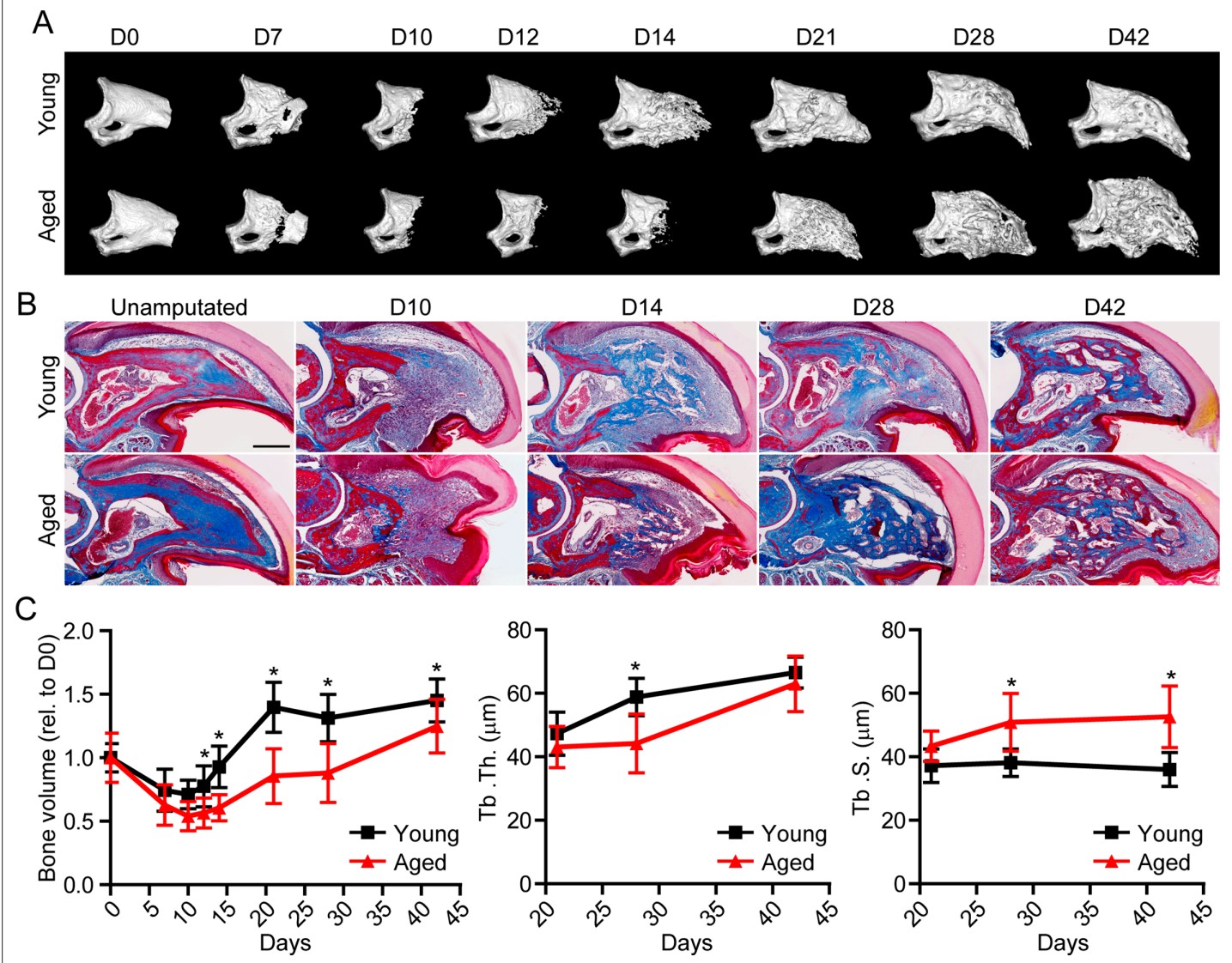

**Figure 2.** Aged mice show impaired regeneration in the digit tip amputation model. (**A**) Radiographic imaging from day 0 to day 42 (D0 to D42) in young and aged mice following amputation of the distal P3. (**B**) Masson's trichrome staining of young and aged mice. (**C**) Micro-computed tomography (micro-CT) quantification of bone volume (relative to D0), trabecular thickness (Tb. Th.), and trabecular separation (Tb. **S**.) of regenerated bone over time. Graphs represent average values ± SD. n=4–14 digits/time point/age. *p<0.05.

analysis to identify genes showing differential expression between the blastema of young and aged regenerating digits (*Supplementary file 5*). Pathway analysis of DEGs showing differential expression in the young or old blastema identified several potentially important factors altered in our impaired regeneration model (*Figure 3A*). First, we identified enrichment in several pathways linked to cell proliferation in the aged blastema. Using *CellCycleScoring* frequently employed in scRNAseq analysis (*Nestorowa et al., 2016*), our data demonstrated a significant increase in S-phase and G2M-phase scoring in the aged blastema (*Figure 3B*), resulting in a significant redistribution of blastema and boundary spots into a proliferative phase (*Figure 3C*). This increased proliferation in aged blastema was confirmed histologically by staining for the proliferation marker PCNA (*Figure 3D*). This increased proliferative gene signature in aged mice could be the result of repeated cell cycling, or from a delay in progression of cells through the cell cycle. Blastema from young and old digits were further interrogated for genes linked to cell cycle activity, including genes encoding cyclins, as well as components of the anaphase promoting complex (*Figure 3E*). These data show a predominant increase in genes

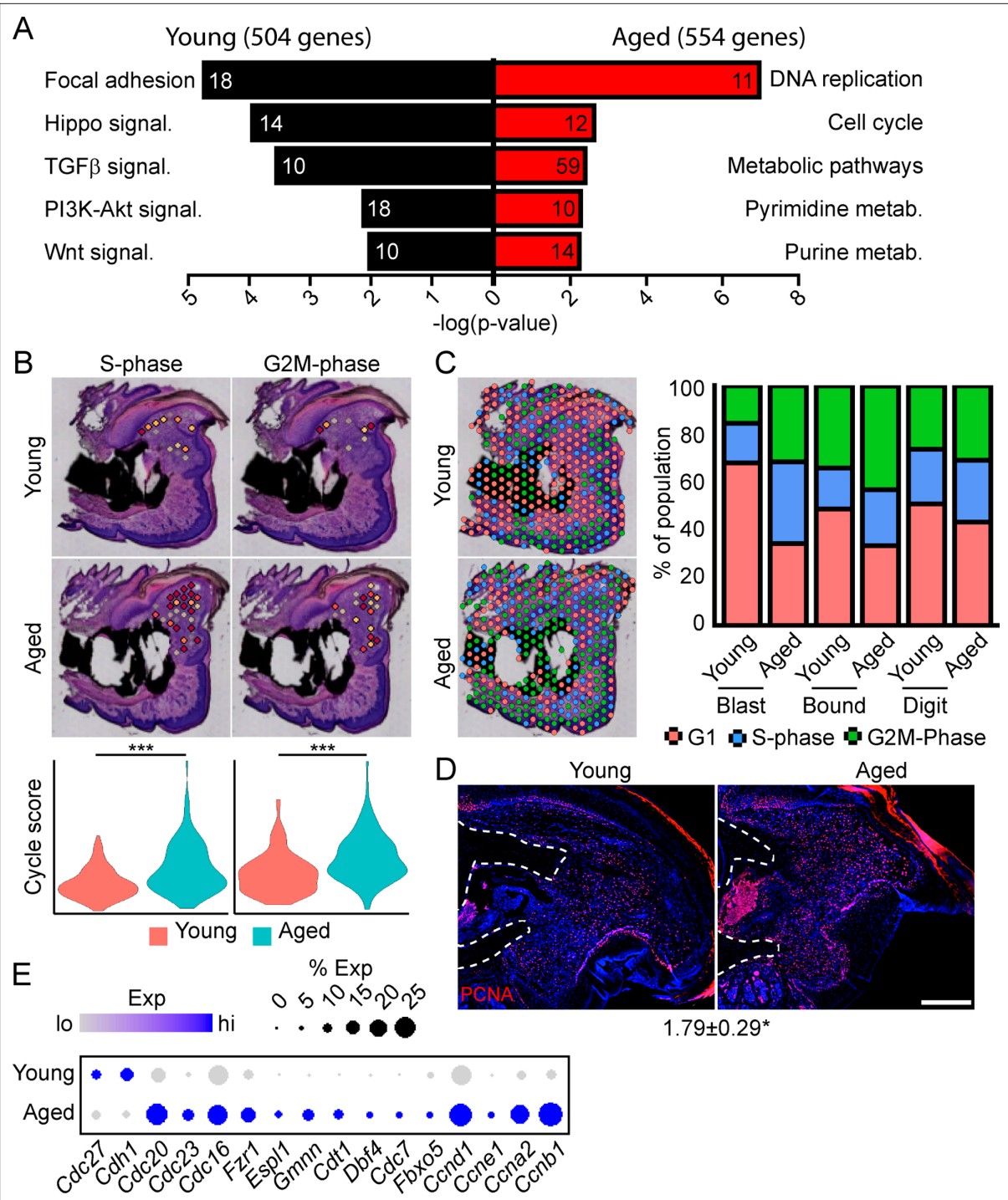

**Figure 3.** Spatial transcriptomics reveals altered cellular metabolism in aged mice. (**A**) Pathway analysis of blastema genes showing differential expression between the young and aged blastema. Number at end of bar denotes the number of significantly regulated genes assigned to each indicated gene ontology (GO term). (**B**) Spatial and violin plots showing S-phase and G2M-phase cell cycle scoring in the blastemas of young and aged mice. (**C**) Cumulative distribution of indicated cell cycle phase assignment for each region of the digit. (**D**) PCNA immunofluorescence in young and aged blastemas. n=3–4. Numbers below image represent the ratio of PCNA-positive nuclei ± SD in aged blastemas relative to young, with values over 1 indicating increased proliferation in aged blastema. Scale bar, 200 μm. (**E**) Dot plot showing relative expression of cell cycle progression genes in young and aged spatial blastema. Values represent average values ± SD. \*\*\*p<0.001, \*p<0.05.

The online version of this article includes the following figure supplement(s) for figure 3:

**Figure supplement 1.** SpatialTime analysis of the young and aged blastema reveals regional specific changes in pathway activation between young and aged mice.

linked to both entry into, and progression through, the cell cycle in aged mice, further supporting an increased cell proliferation in the regenerating blastema of aged mice.

To further interrogate which pathways are alternately regulated in young vs. aged mice, and the spatial patterning of the pathway expression, we conducted a modified SpatialTime analysis, a technique recently developed for analysis to look at signaling gradients within spatial transcriptomic data (*Tower et al., 2021*) (see Materials and methods) (*Figure 3—figure supplement 1A*). Briefly, a reference line is generated along the residual P3 bone stump. Next, the minimum distance between each blastema spatial spot and this reference line is calculated. Finally, the distance of each spot from the reference line is scaled from values of 0 (most proximal) to 1 (most distal). To determine the level of pathway activation, we generated gene lists from curated KEGG pathways, followed by module scoring. Whole blastema analysis shows notable reductions in the Hippo, WNT, and TGFβ pathways in aged mice, with SpatialTime analyses suggesting that both Hippo and WNT signaling are preferentially decreased distally in aged mice, while TGFβ signaling was decreased proximally, but had comparable expression distally (*Figure 3—figure supplement 1B*).

Beyond morphogenetic pathways, analysis of the aged blastema also showed enrichment for DEGs linked to cellular bioenergetics, and that these metabolic pathways were divergently regulated in young and aged mice (*Figure 3A*). Owing to the important role of cellular metabolism in cell proliferation, fate determination and function (*Lee et al., 2020*; *Loeffler et al., 2018*; *Lee et al., 2017*; *Guntur et al., 2014*; *Riddle and Clemens, 2017*), as well as the known role of aging in cellular metabolism (*Lesnefsky and Hoppel, 2006*; *Hadjiargyrou and O'Keefe, 2014*), we focused our attention on differential metabolic programming between young and aged mice. Metabolic analysis showed increased glycolysis and oxidative phosphorylation (OxPhos) in the blastema of aged mice, resulting in an increased overall energetic response (cumulative glycolysis and OxPhos scoring) (*Figure 4A*). This increased bioenergetic signature was specific for glucose metabolism, as scoring of fatty acid metabolism failed to show any significant increase. A more detailed analysis of respiratory energy metabolism suggests that aging upregulates the TCA cycle and complexes IV and V of the electron transport chain, with no significant change in mitophagy or mitochondrial biogenesis (*Figure 4—figure supplement 1*). Applying our SpatialTime technique, we observed that glycolysis is preferentially upregulated within the proximal blastema of aged digits and OxPhos increased in the distal blastema (*Figure 4B*).

To validate differences in cell metabolism, we analyzed excised blastemas from young and aged digits and evaluated cell metabolism using Seahorse analysis (*Figure 4C*). Oxygen consumption rates (OCR) associated with basal respiration ($p<0.05$), ATP ($p<0.05$), maximal respiration ($p<0.05$), and non-mitochondrial respiration ($p<0.05$) were higher in the aged blastema-derived cells. The extracellular acidification rate (ECAR) (Mito Stress Test) was also significantly higher after each stressor injection, suggesting increased glycolytic rates. To address potential mitochondrial-driven $CO_2$ acidification in the ECAR analysis, we also evaluated the glycolysis-mediated protein efflux rate. The Glycolytic Rate Assay (Seahorse analysis) confirmed a significant increase in the capacity for aged blastema cells to respond to glucose by upregulating compensatory glycolysis ($p<0.05$). Combined, these data support an elevated level of cell metabolism within the blastema-derived cells of aged mice.

## Aged blastemas demonstrate prolonged intracellular hypoxia and increased vascularization

We have previously shown that intracellular hypoxia is an integral component of digit regeneration (*Sammarco et al., 2014*; *Simkin et al., 2015*). Owing to our observed increase in bioenergetic transcripts in vivo and increased oxygen consumption in vitro, we next sought to determine the potential effects of increased metabolism on hypoxia within the regenerating digit. Spatial transcriptomics module scoring (*Figure 5A*), as well as confirmatory histological labeling using hypoxyprobe (*Figure 5B*), both confirm elevated levels of hypoxia within the blastema of aged mice. One prominent pathway highly regulated by hypoxia is vascular endothelial growth factor (VEGF) signaling (*Shweiki et al., 1992*), a pathway also known to heavily influence bone development and repair (*Hu and Olsen, 2016*; *Schipani et al., 2009*). To determine whether this shift in hypoxic signaling was associated with shifts in downstream levels of VEGF signaling, and subsequent vascularization, we quantified each of these parameters in our spatial data using module scoring. Both VEGF signaling (*Figure 5C*) and vascular markers (*Figure 5D*) showed significant elevations within the blastema of

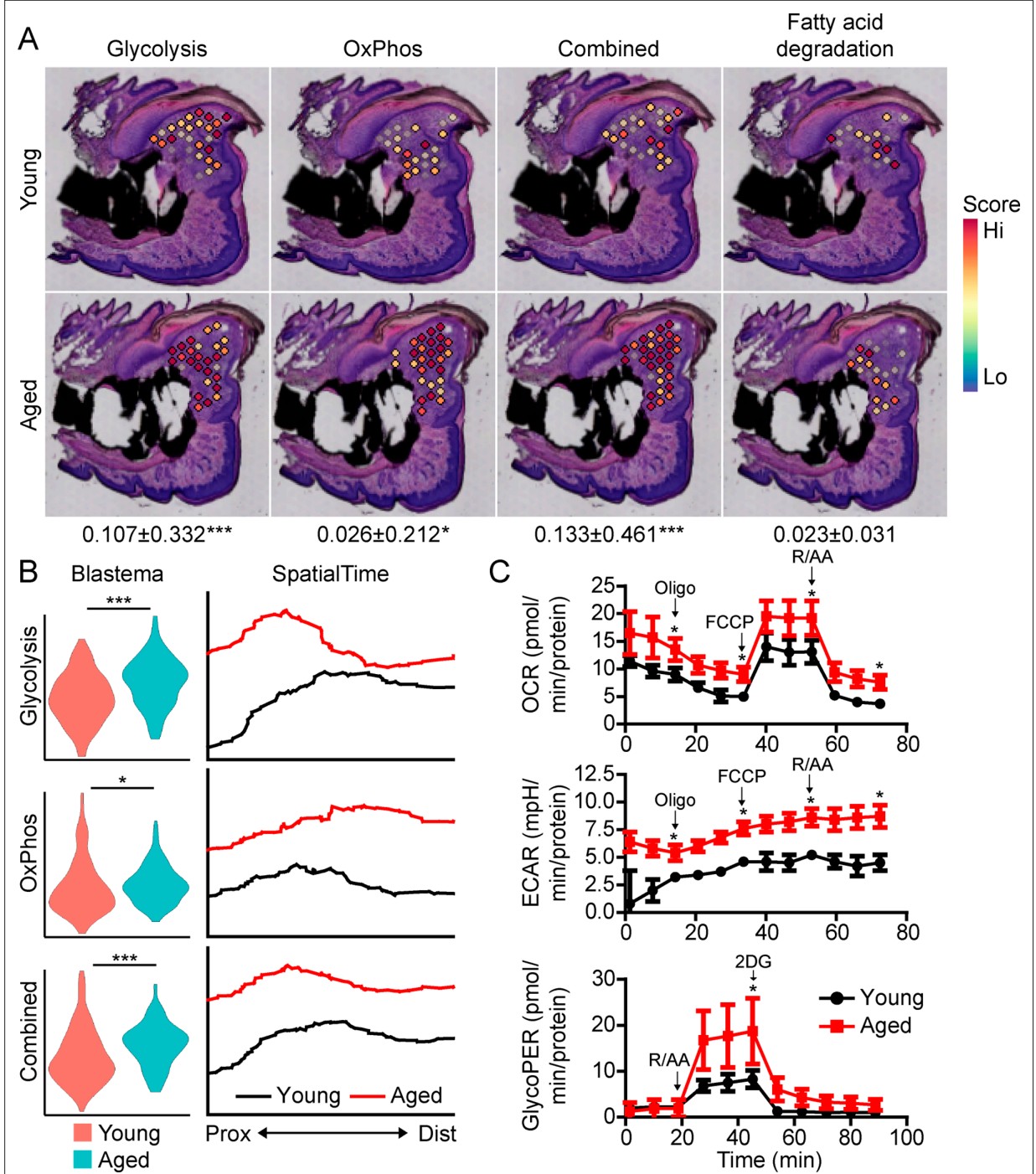

**Figure 4.** Aged blastemas have increased bioenergetics. (**A**) Module scoring for cell metabolism pathways within young and aged blastemas. Combined is the cumulative module scoring values of glycolysis and oxidative phosphorylation (OxPhos). Values below images indicate difference in spatial spot module score within pooled blastema of aged vs. young mice ± SD, with values above 0 indicating increased activation in aged blastema. (**B**) Violin (left) and SpatialTime plot showing metabolic module score. Prox, proximal; Dist, distal. (**C**) Seahorse Mito Stress Test (oxygen consumption rate, OCR; extracellular acidification rate, ECAR) and Glycolytic Rate Assay (glycolygic proton efflux rate, GlycoPER) of dissected blastema from young and aged mice (Oligo, oligomycin; R/AA, rotenone/antimycin A). n=8–9 mice. Graphs represent average values ± SEM. *p<0.05, ***p<0.001.

The online version of this article includes the following figure supplement(s) for figure 4:

**Figure supplement 1.** Metabolic pathway-specific changes within the blastema of young and aged mice.

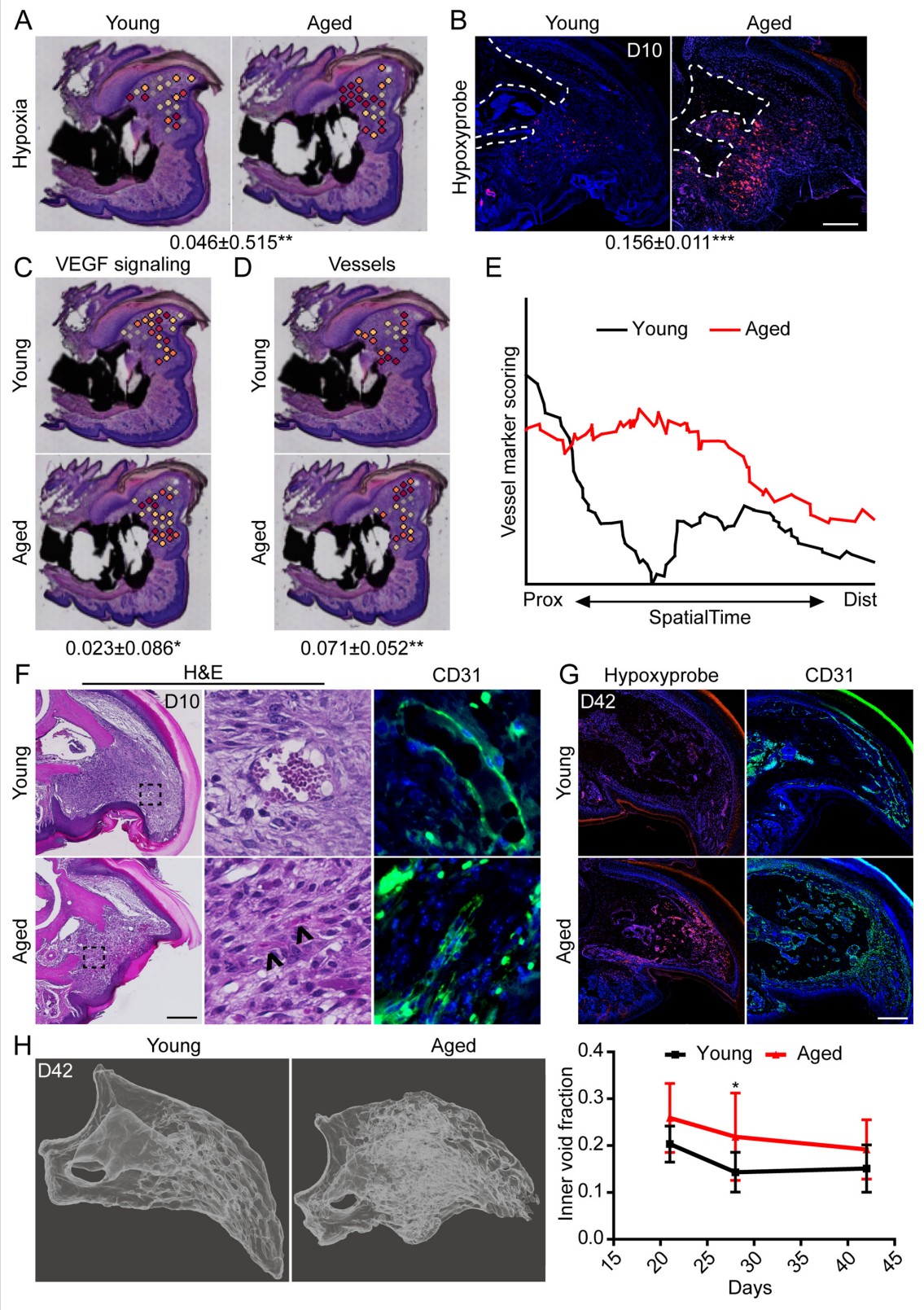

**Figure 5.** Increased intracellular hypoxia and vascularization in aged mice. (**A**) Spatial module scoring of hypoxia-related transcripts. (**B**) Hypoxyprobe staining in young and aged blastema. Scale bar, 100 µm. Dotted lines indicate the P3 cortical bone stump. (**C**) Spatial module scoring of vascular endothelial growth factor (VEGF) signaling. (**D**) Spatial module scoring of vascular markers. Values below images indicate difference in spatial spot module score or hypoxyprobe fluorescent intensity within pooled blastema of aged minus young mice ± SD, with values above 0 indicating increased

*Figure 5 continued on next page*

*Figure 5 continued*

activation in aged blastema. (**E**) SpatialTime analysis of the vascular module score from the proximal (residual bone stump) to distal blastema tip in young and aged mice. (**F**) Hematoxylin and eosin (H&E) staining and immunofluorescent imaging of CD31 showing increased vascularization within the blastema of aged mice at day 10. Scale bar, 100 μm. Arrow heads denote immature vessels. (**G**) Hypoxyprobe and CD31 staining of digits at day 42. Scale bar, 250 μm. (**H**) Micro-computed tomography (micro-CT)-based inner void calculations within the regenerated digit. Graphs represent average values ± SD. n=4–6 samples/condition. *p<0.05, **p<0.01, ***p<0.001.

aged mice. SpatialTime analysis confirmed that aged mice demonstrated increased gene expression for vessel markers concentrated predominantly in the proximal portion of the blastema (*Figure 5E*). This is in contrast to blastemas from young mice that showed minimal expression of vessel markers concentrated mostly in the distal tip (*Figure 5E*). Histological analyses confirmed that while large, CD31$^+$ vessels were predominantly seen in the distal tip of the blastema in young mice, additional small and/or immature vessels were detected throughout the blastemas of aged mice (*Figure 5F*). Interestingly, this increased in intracellular hypoxia and altered vessel patterning persisted throughout the healing process in aged mice, with significantly greater hypoxyprobe labeling and CD31$^+$ blood vessels present throughout the regenerating digit at D42 (*Figure 5G*). These changes in vascularization were accompanied by altered regenerative bone architecture characterized by multidirectional and disorganized void space in the aged mice and an overall increased inner void fraction (*Figure 5H*).

### The metabolite OAA enhances skeletal regeneration in aged mice

Our data show that there is an increase in both glycolytic and oxidative metabolism pathways in the aged blastema, suggesting an increased energy demand. We hypothesized that unmet energy needs may underpin age-dependent metabolic increases, and that supplementing these cells with a substrate to off-set these increased requirements would enhance regeneration. Lee et al. recently demonstrated that active glycolysis in the mature osteoblast relies heavily on the malate-aspartate shuttle (*Lee et al., 2020*). This shuttle serves to move OAA into the mitochondria by first converting it to malate. Once inside the mitochondrial membrane malate is oxidized back to OAA again, generating NADH directly in the mitochondria for use in the electron transport chain (*Lee et al., 2020*). We hypothesized that supplementing with intermediates from the malate-aspartate shuttle would support this metabolic pathway and help meet the additional bioenergetic demand demonstrated by both spatial and Seahorse data. To this end we utilized the anapleurotic metabolite OAA, which has been shown to promote brain mitochondrial biogenesis and increase ATP production in nerves via systemic application (*Wilkins et al., 2016*; *Wilkins et al., 2014*).

In vitro, Seahorse analysis on aged blastemas using the Mito Stress Test revealed a significant decrease in maximal respiration after the addition of OAA, with no significant change in ECAR or glycolytic flux (*Figure 6A*). To determine whether this OAA-driven decrease in maximal respiration within the blastema translates into changes in cell function, we administered OAA daily to aged mice starting at the blastema phase (D10). 3D micro-CT image analysis of skeletal regeneration after OAA treatment revealed that while only slight increases in regenerative bone parameters were observed by day 21, significant increases in regenerated bone volume and thickness were observed in response to daily OAA treatment from D10 to D28 (*Figure 6B and C*). 2D analysis of the regenerated bone at day 28 post amputation, oriented along the axis of healing from the proximal to distal end, demonstrated that this increase in bone is predominantly within the first 0.4 mm of the bone stump, rather than the distal end of the bone (*Figure 6D*). These data indicate that treatment with the metabolite OAA can enhance regeneration in aged mice.

### OAA-induced regeneration is associated with enhanced proliferation, bioenergetics, and BMP/WNT signaling

To determine OAA's mechanism of action, we conducted spatial transcriptomics on aged samples treated with OAA. To identify OAA-dependent bone morphogenetic and cell function pathways, we evaluated aged samples during early bone formation phase (D21), treating with OAA daily from day 10 to 21, segmenting the area into a fibroblast and bone region (*Figure 7A*). Cell cycle analysis revealed that OAA further increased the number of cells in S-phase and G2M-phase scoring in both the fibroblast and bone areas of the regenerating digit (*Figure 7B*). This increase in proliferation was confirmed histologically by staining for the proliferation marker PCNA (*Figure 7C*). Given that

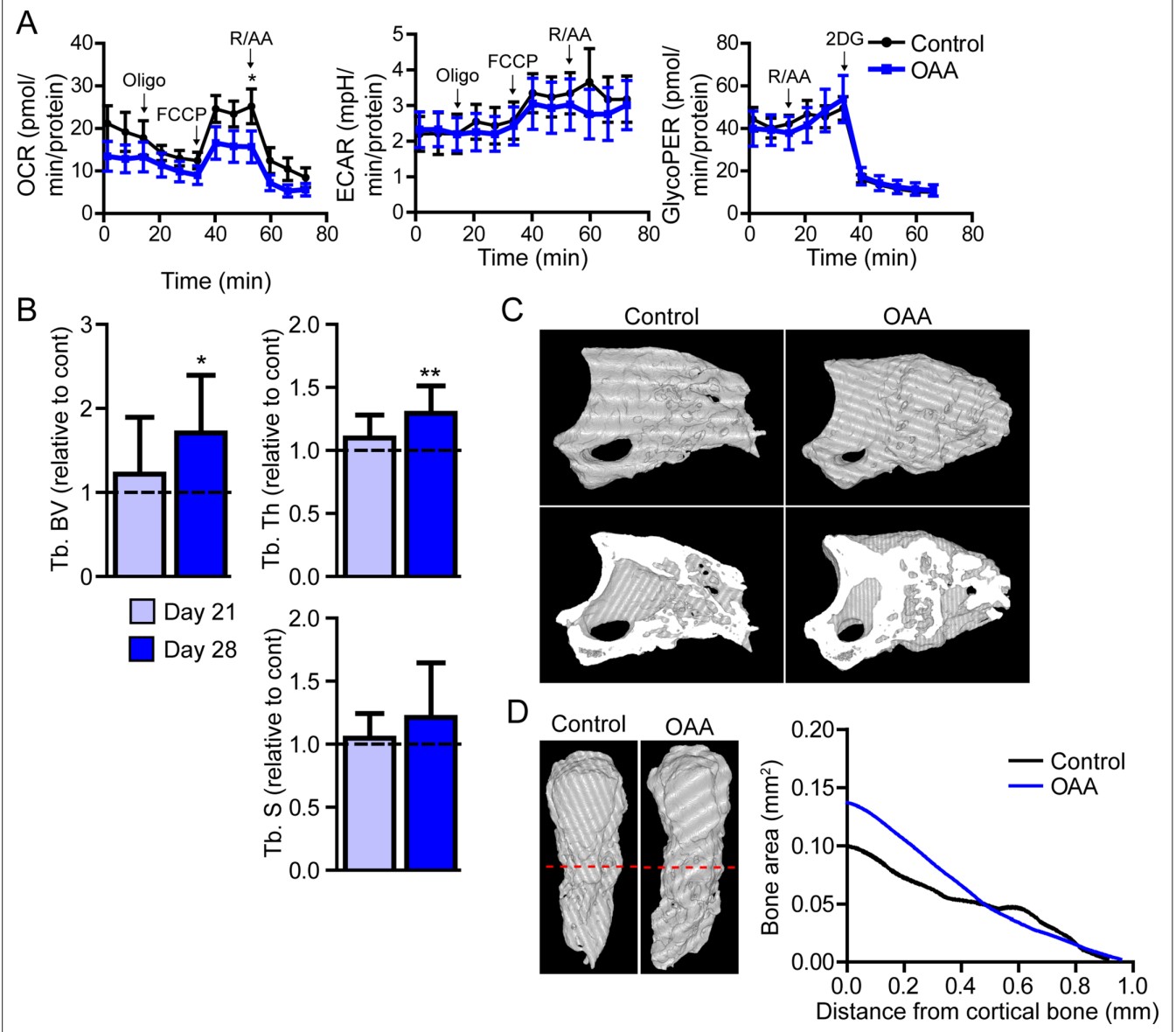

**Figure 6.** Treatment with oxaloacetate (OAA) enhances regeneration in aged mice. (**A**) Seahorse Mito Stress Test (oxygen consumption rate, OCR; extracellular acidification rate, ECAR) and Glycolytic Rate Assay (glycolygic proton efflux rate, GlycoPER) of dissected blastema from aged mice treated with oxaloacetate (OAA) (Oligo, oligomycin; R/AA, rotenone/antimycin A). (**B**) Micro-computed tomography (micro-CT) quantification of trabecular bone volume (Tb. BV.), trabecular thickness (Tb. Th.), and trabecular separation (Tb. **S**.) of regenerated OAA treated bone at day 21 and day 28 post amputation, relative to time-matched, saline-treated controls. n=11–19 digits/group. Graphs represent average values ± SEM. *p<0.05, **p<0.01, ***p<0.001. (**C**) Representative whole (top) or bisected (bottom) day 28 digits from saline control or OAA-treated mice. (**D**) 2D micro-CT analysis of regenerated bone from the remaining P3 cortical bone stump extending distally from aged saline control and OAA-treated mice.

administration of OAA enhances bone volume and thickness, but not spacing, we investigated the impact of OAA on vascularity and hypoxia-driven pathways. Spatial transcriptomics module scoring (*Figure 7D*), as well as confirmatory histological labeling using hypoxyprobe (*Figure 7E*), both showed that OAA treatment increased hypoxia expression levels in the fibroblast area and reduced hypoxia expression levels in bone area of the D21 aged digits. These values were inversely correlated with VEGF signaling, while vascular markers were found to be decreased in both regions (*Figure 7F*). Treatment with OAA was also found to affect cell metabolism, with OAA administration resulting in increased glycolysis in both the fibroblast and bone areas, while OxPhos was found to be increased only within the fibroblast region (*Figure 7G*).

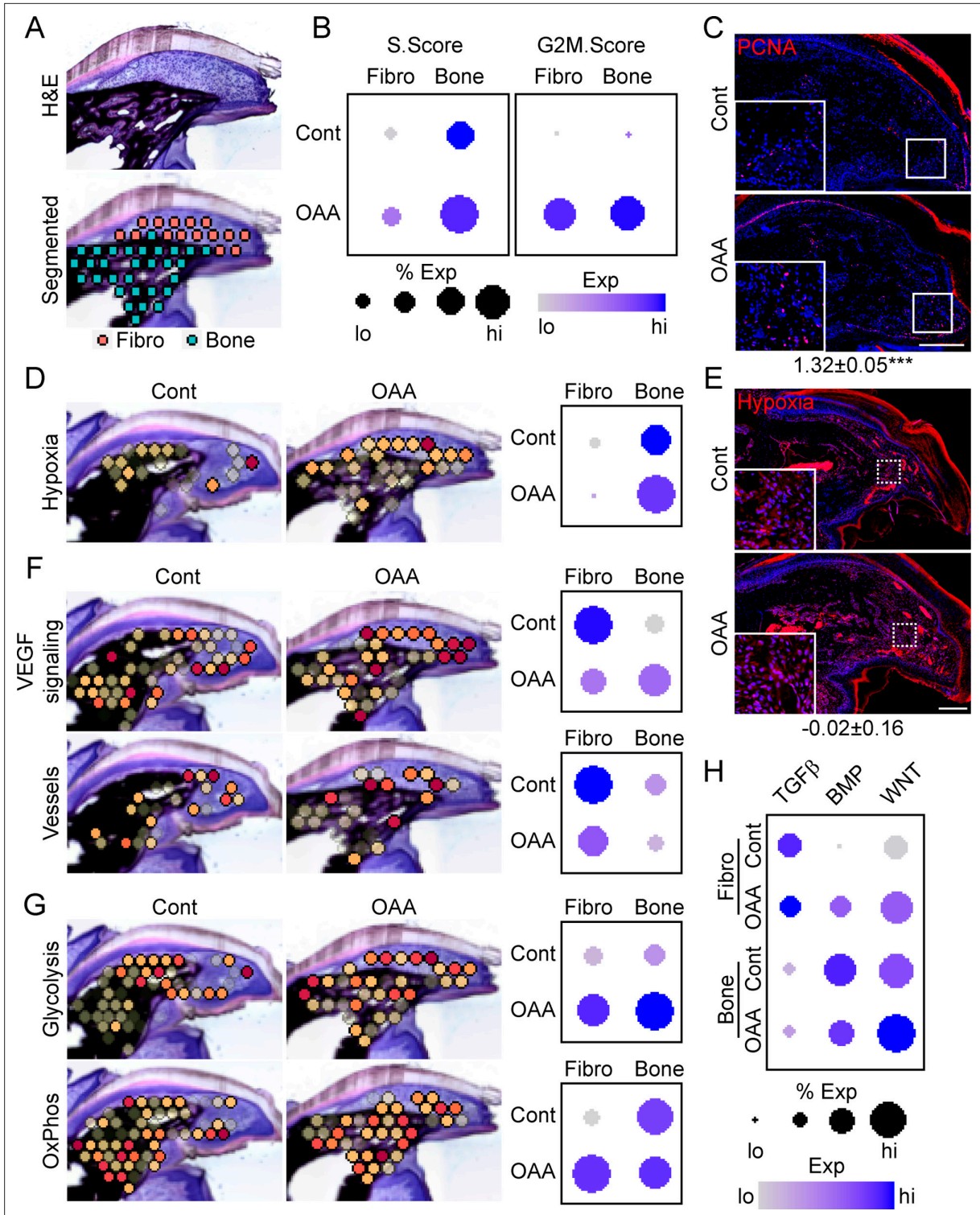

**Figure 7.** Oxaloacetate (OAA) treatment promotes proliferation and WNT signaling. (**A**) Aged mice treated daily with OAA from D10 were sacrificed at D21 and subjected to spatial transcriptomics and the regenerated digit divided into fibroblast (fibro) and bone areas. (**B**) S- and G2M-phase scoring of aged mice treated with OAA or saline controls (Cont). (**C**) PCNA immunofluorescence in OAA-treated aged mice vs. saline controls (**D21**). n=3. Numbers below image represent the ratio of PCNA-positive nuclei ± SD in aged mice treated with OAA relative to saline control, with values over 1 indicating increased proliferation in aged blastema. Scale bar, 200 µm. (**D**) Spatial and dot plots showing module scoring of hypoxia pathway activation. (**E**) Hypoxia as assessed by Hypoxyprobe immunofluorescence (**D21**). Values below images indicate difference in Hypoxyprobe fluorescent intensity within the regenerated digits of aged mice treated with OAA or saline control ± SD, with values below 0 indicating decreased activation in OAA-treated

*Figure 7 continued on next page*

*Figure 7 continued*

digits. Scale bar, 200 µm. (**F**) Spatial and dot plots showing module scoring of vascular endothelial growth factor (VEGF) signaling activation and vessel markers. (**G**) Spatial and dot plots showing module scoring of glycolysis and oxidative phosphorylation (OxPhos) activation. (**H**) Dot plots showing module scoring of transforming growth factor beta (TGFβ), BMP, and WNT pathway activation. n=4–7 digits per treatment group.

Pathway analysis on both the fibroblast and bone DEGs (*Supplementary file 6*) indicated that TGFβ, BMP, and WNT are potentially regulated in response to OAA treatment. To determine the effects of OAA or morphogenetic signaling, we assessed modular scoring of pathway activation (*Figure 7H*). WNT signaling was found to be upregulated in both the fibroblast and bone regions, while BMP signaling was increased in the fibroblast region and decreased in the bone region. Together, these data suggest that administration of OAA enhances regeneration by increasing proliferation and promoting differentiation through regulation of morphogenetic pathways.

## Discussion

In this study, we report an important role for cell metabolism in regulating blastema function and regenerative potential. To identify potential discerning pathways differentially regulated between regenerating and non-regenerating tissue, we applied spatial transcriptomic analysis in an aged mouse model. Our results show impaired digit regeneration in aged mice, associated with increased proliferation and decreased bone building, as well as differential proximal-distal patterning of glycolysis and OxPhos. These increased metabolic demands in aged mice are associated with increased intracellular hypoxia, VEGF signaling, and vascularization. Finally, we demonstrate that administration of the metabolite OAA can partially rescue this age-dependent decline in regeneration.

### Age-dependent delays in regeneration

While extensive work has been conducted to characterize the blastema in multiple animal models and regenerating conditions, it was only recently that a comprehensive transcriptional profile was available for blastema fibroblasts (*Storer et al., 2020*; *Johnson et al., 2020*). Using scRNAseq, these studies made significant contributions to our understanding of blastema cell subtypes and their transition throughout the regenerative process. Single-cell analysis, however, is unable to provide spatial information, making it difficult to place the blastema in the context of its neighboring tissues. Spatial transcriptomics, shown here for the first time in digit regeneration, allows transcriptional insight into the histologically defined blastema in the context of its spatial position within the regenerating digit. Using this technique, we identified a blastema transcriptional profile delineating this structure in not only our own spatial data, but also within previously published scRNAseq studies (*Storer et al., 2020*; *Johnson et al., 2020*). The strong correlation between these data sets suggests a high degree of conservation between spatial transcriptomic and scRNAseq data, and thus, opens up exciting new avenues for future research across platforms.

Even for highly regenerative animals like the Mexican axolotl (*McCusker et al., 2015*), age is associated with a decline in regenerative capacity (*Seifert and Voss, 2013*; *Vieira et al., 2020*; *McCusker and Gardiner, 2011*). Similarly, in humans, while regenerative capacity of the digit tips persists into adulthood, fidelity of regeneration appears to decline with age (*Muneoka et al., 2008*). We show here that age is associated with a delay in the switch from bone catabolism to bone anabolism, and that the final regenerate has significantly increased trabecular spaces, reduced volume, and overall impaired bone patterning, suggesting a defect in bone anabolism in aged mice. This delay is accompanied by increased proliferation, which may indicate that delays in bone regeneration may be linked to delays in cell cycle progression in the aged blastema. Increased proliferation in the aged blastema may also explain the increases in cell metabolism, since proliferation would require both energy and biosynthesis. Most studies agree that formation of the blastema relies on, at least in part, the dedifferentiation of terminal cells within the remaining digit (*Storer et al., 2020*; *Johnson et al., 2020*; *Lehoczky et al., 2011*; *Rinkevich et al., 2011*). In contrast, proliferation is thought to play a relatively less critical role in blastema formation (*Storer et al., 2020*; *Johnson et al., 2020*). While our studies are not able to determine the relative contribution of dedifferentiation to blastema formation in young and aged mice due to the lack of reliable markers, this increased proliferation may point to a potential loss of plasticity, and the subsequent loss of dedifferentiation within aged cells. Another possible explanation

for the delay is that the increased intracellular hypoxia may be suppressing osteoblast differentiation, a concept demonstrated in both osteoblasts (*Utting et al., 2006*) and in the blastema (*Sammarco et al., 2014*). However, this runs counter to our OAA-treated mice which showed increased bone formation without altering intracellular hypoxia.

## Mechanistic changes in digit regeneration during aging

VEGF expression is intimately linked to bone repair and regeneration (*Hu and Olsen, 2016*). Our initial analysis of the aged blastema showed increased hypoxia, VEGF signaling, and vascular markers. Intramembranous ossification relies on hypoxia-induced VEGF expression (*Wang et al., 2007*) to increase formation of vasculature and mobilize bone progenitors (*Hu and Olsen, 2016*; *Matsubara et al., 2012*). However, excess VEGF is also known to inhibit the function of pericytes (*Greenberg et al., 2008*), leading to formation of immature blood vessels and uncoupling of angiogenesis and osteogenesis. In addition, excess VEGF signaling has also been shown to inhibit digit regeneration in neonates (*Yu et al., 2014*). Together, these data suggest that excess VEGF production driven by hypoxia in the blastema of aged mice may contribute to altered bone architecture and impede regeneration in aged mice.

Despite the increased vascularization, the regenerating digits from aged mice still demonstrated increased intracellular hypoxia. However, in vitro studies (*Kurokawa et al., 2015*; *Prior et al., 2014*) suggest that this hypoxia may be directly driven by oxygen consumption by the electron transfer chain in mitochondria (*Friedman and Nunnari, 2014*) and continues to be an ongoing area of research (*Agathocleous and Harris, 2013*). A recent study by *Yao et al., 2019*, confirmed this same phenomenon in bone in vivo, noting that intracellular hypoxia modulates epigenetic modifications (*Batie et al., 2019*; *Chakraborty et al., 2019*; *Gallipoli and Huntly, 2019*), as well as gene and protein expression (*Semenza, 2017*). Functionally, Yao et al. demonstrate that the use of non-oxidative glycolysis protects hypoxic tissues from anoxia by regulating intracellular hypoxia during development. Lee et al. fortify this finding by showing that reduced oxygen consumption is a hallmark of osteoblasts (*Lee et al., 2020*). Guided by our results, and in concert with these published findings, we propose that upregulation of OxPhos during skeletal regeneration in aged mice facilitates hypoxia-driven vasculature that ultimately disrupts bone architecture.

Bioenergetics were some of the major pathways altered during aging regeneration. Our spatial transcriptomic data and Seahorse data show increased glycolysis and OxPhos in aged D10 blastema fibroblasts when compared to the young blastema. We sought to address the unmet energy needs of the aged blastema with OAA, given that mature osteoblasts rely heavily on the malate-aspartate shuttle (*Lee et al., 2020*), and that OAA provides a source of both NADH and biomass (*Stephanopoulos et al., 1998*). Phenotypically, OAA partially resolved the aged phenotype by increasing bone volume and thickness, but did not decrease trabecular spacing.

Closer investigation of the transcriptomic changes at the early bone formation stage (D21) revealed that OAA had differential effects on the fibroblasts and bone regions at D21. Within the fibroblast region, OAA increased glycolysis and OxPhos, proliferation, and hypoxia, and reduced VEGF signaling and vascular markers. Within the bone region, OAA demonstrated increased glycolysis, but not OxPhos, and was accompanied by an increase in proliferation and VEGF signaling, but not vascular markers or hypoxia. These OAA-dependent changes were accompanied by increases in WNT pathway expression in both fibroblasts and bone, which may be driving these anabolic bone processes. The regulatory axis between WNT and cell metabolism has only recently begun to be explored in bone, and while evidence has grown showing the impact of WNT on cell metabolism (*Karner and Long, 2017*; *Karner et al., 2015*; *Kobayashi et al., 2016*), data is scarce in understanding the converse relationship where cell metabolism modulates WNT signaling (*Costa et al., 2019*; *Delgado-Deida et al., 2020*; *Risha et al., 2021*; *Zhang et al., 2019*). WNT is a central regulator of bone repair and regeneration, and is known to differentially affect various cell lineages (*Houschyar et al., 2018*), which may partially explain the divergent effects in fibroblasts and bone. Further work is required to determine whether OAA-mediated rescue requires WNT signaling activation, and whether WNT is directly or indirectly regulated by OAA-induced cell metabolism.

While OAA-dependent increases in glycolysis and OxPhos could be attributed to the differential effects of WNT signaling, it's also important to consider the critical signaling role of OxPhos-mediated ROS generation in limb regeneration. ROS production after tail amputation in *Xenopus* is necessary

for regeneration, possibly through WNT signaling and proliferation (*Love et al., 2013*), and leptin and melanocortin 4 receptor, key molecules regulating energy balance, are critical in tadpole tail regeneration (*Love et al., 2011*) and digit regeneration (*Kang et al., 2016*; *Zhang et al., 2018*). While we chose OAA partially because it supplemented the elevated cell metabolism levels seen at D10, these increased metabolism levels may be an inherent and essential part of limb regeneration, with OAA potentially providing building blocks for cell growth and expansion, further increasing cell metabolism. Thus, it may be misguided to think of promoting regeneration by reversing a molecular phenotype, and may, instead, be an exercise in determining what is needed metabolically and providing that substrate within the existing framework. Future studies with treatment of OAA or other metabolites, combined with the ability to spatially analyze signaling pathways and cell metabolism, may identify molecular targets to better promote skeletal regeneration and bone repair and further refine the timing of these treatments.

## Considerations for future studies

While spatial transcriptomics offers the unique ability to discern gene patterning in tissue across the entire transcriptome, it is important to note the limitations of this approach. Optimal permeabilization is a balance between RNA release and excess diffusion that results in loss of spatial specificity. This balance does not promote the complete recovery of mRNA within the overlaid tissue section, restricting the capture efficiency of the transcriptome. Analytically, while our results suggest that the level of transcriptional information obtained per spatial spot is consistent with typical drop-seq-based single-cell experiments, the latter relies on a large sample size to help computationally 'fill in' dropout transcriptional information with that of its co-clustering neighbors. Our spatial information likely reflects a similar loss of transcriptional information but without the availability of thousands of spots to compensate. As such, individual gene signatures must be considered with this limitation in mind. To overcome this drawback, we made use of module scoring, which assess the expression distribution of a list of genes (relative to a random background list). Another limitation is the low resolution, with each spatial spot representing the transcriptome from multiple cells. As such, we are unable to distinguish cell heterogeneity within close proximity. This is evident by a lack of *Ptprc* expression within the regenerating digit, in contrast to previous works which have demonstrated the essential presence of, for instance, macrophages in the regenerative process (*Simkin et al., 2017*). This lack of *Ptprc* expression likely reflects the fact that immune cells are scattered among other cell types, resulting in 'masking' of their individual transcriptome. In addition, we make use of recently developed SpatialTime analysis (*Tower et al., 2021*) which allows concatenation of multiple samples to reflect gene expression changes across an 'average' blastema. These tools significantly mitigate inherent limitations within each individual spatial spot.

This study addresses a direct comparison of young and aged digit regeneration where our data support an age-dependent delay in bone regeneration. To address differences in gene expression, we analyze both the aged and young blastema 10 days after amputation. While time course studies suggest regeneration delays occur at the level of osteogenic commitment and mineralization, using day 10 for both ages may capture differences that are merely representative of a chronological shift. Our studies in hypoxia and proliferation suggest that these gene expression differences are not simply a delay, however it is important to take this possibility into consideration. These data pave the way for additional studies that will hopefully illuminate the temporal activation pathways throughout the entire regenerative process.

While our data demonstrate that OAA can reverse aging-induced declines in regeneration, the exact mechanism of rescue remains to be elucidated. In our studies, OAA was administered throughout the blastema and bone formation phases. While pro-osteogenic transcriptional changes observed at D21, prior to any significant changes in bone architecture, suggests that OAA-mediated rescue, at least in part, functions upstream of the mineralization process to promote cell fate determination, we cannot rule out the possibility that OAA may act at the level of promoting osteoblast mineralization during bone formation and late-stage remodeling. In light of the differential effects of OAA on the fibroblast and bone areas of the digit, future work is needed to address the specific effects of OAA on the blastema and bone formation phases separately.

## Conclusion

This study provides spatial registration of blastema gene expression differences in aged and young mice, and confirms prior expression profiles of the blastema (*Storer et al., 2020*; *Johnson et al.,*

*2020*). Our data suggest that cell metabolism is altered during aging, resulting in impaired regeneration, and that while administration of the metabolite OAA is able to enhance bone formation, it is only able to partially rescue the phenotype. Furthermore, our data show that OAA-dependent regeneration is driven, at least in part, by upregulation of canonical WNT signaling. Future studies will focus on further dissecting the molecular mechanisms link metabolite signaling to these pathways.

# Materials and methods

## Key resources table

| Reagent type (species) or resource | Designation | Source or reference | Identifiers | Additional information |
|---|---|---|---|---|
| Strain, strain background (*Mus musculus*) | CD1 | Charles River Laboratories | Strain 022 | Female and male, 6–22 months of age RRID:IMSR_CRL:022 |
| Chemical compound, drug | Oxaloacetate | Sigma | 07753 | Oxaloacetic acid |
| Other | Z-Fix | Anatech | 5701ZF | Fixative |
| Other | Decal I | Surgipath | | Decalcifier |
| Other | Blocking solution | Thermo | 27515 | Blocking solution |
| Commercial assay, kit | Masson's Trichrome Kit | Poly Scientific | K037 | Stain kit |
| Commercial assay, kit | Tyramide signal amplification | Invitrogen | T20924 | |
| Commercial assay, kit | Hypoxyprobe Plus Kit | Hypoxyprobe | HP2 | Stain kit |
| Commercial assay, kit | Mito Stress Test Kit | Agilent Seahorse | 103015-100 | |
| Commercial assay, kit | Glycolytic Rate Assay Kit | Agilent Seahorse | 103344-100 | |
| Software, algorithm | GEN5IPRIME V3.05.11 | Biotek | | |
| Software, algorithm | GraphPad Prism 9 | GraphPad | | |
| Software, algorithm | CTAn | Bruker | | |
| Software, algorithm | NRECON | Bruker | | |
| Software, algorithm | CTVox | Bruker | | |
| Antibody | (Rabbit Monoclonal) Anti-CD31 antibody | Abcam | ab182981 | (1:100) |
| Antibody | (Mouse Monoclonal) Anti-PCNA antibody | Abcam | ab29 | RRID:AB_303394 (1:200) |
| Software, algorithm | CellRanger | Version 6 | | 10× Genomics |
| Software, algorithm | Seurat | Version 3 | R package | *Stuart et al., 2019* |
| Software, algorithm | ggpubr | Version 0.4.0 | R package | Kassambara, STHDA July 2016 |

## Experimental model and subject details

### Mice

All experiments were performed in accordance with the standard operating procedures approved by the Institutional Animal Care and Use Committee of Tulane University Health Sciences Center (Protocol #1483). In all cases mice had free access to low-fluorescence rodent chow and water in a 12 hr dark-light cycle room. For all studies, mice of either sex were used and mice were randomly allocated to experimental groups. The ages of mice used for experiments ranged from 6 to 22 months. Experimental endpoints are noted in the figure legend. Wild-type CD1 mice were purchased from Charles River Laboratories (strain 022). Four digits from four separate mice were used for spatial transcriptomics for each age group (eight total hindlimb digits per group).

## Method details

### Amputations and OAA treatments

Adult young (6–7 months) and aged (18–22 months) female and male CD1 wild-type (002) mice were purchased from Charles River Laboratories. Mice were anesthetized with 1–5% isoflurane gas with continuous inhalation. The second and fourth digits of both hindlimbs were amputated at the P3 distal level as described previously (*Sammarco et al., 2015*; *Busse et al., 2019*) and regenerating

digits were collected at days 0, 7, 10, 14, 21, 28, and 42 for analysis. The third digit was used as an unamputated control. OAA (Sigma, 07753) was dissolved in phosphate-buffered saline (PBS) and pH-adjusted with NaOH to 7.0 to allow the OAA solution to drift toward alkaline levels as described by *Wilkins et al., 2016*. Mice were treated daily with 0.5 g/kg OAA in 200 µL via i.p. injection based on our own studies that showed aged mice routinely survived repeated doses at 0.5 g/kg but not 2 g/kg as described in the previous paper (*Wilkins et al., 2016*). The vehicle-treated group received PBS. Mice were dosed from day 10 until day 28.

## Tissue collection, fluorescence, and histology

Digits were fixed overnight in zinc-buffered formalin (Z-fix; Anatech). Bone was decalcified for 48 hr in a formic-acid-based decalcifier (Decal I, Surgipath). Once decalcified, all samples were processed for paraffin embedding. Sections were stained with Masson's trichrome (Poly Scientific, K037-16OZ) and mounted using permanent mounting medium (Thermo, SP15-100). Immunofluorescent staining was performed on deparaffinized and rehydrated sections. Antigen retrieval was performed using heated antigen retrieval solution (Vector, H-3300) and allowed to cool to room temperature prior to blocking with blocking solution (Thermo, 37515). Sections were incubated with primary antibodies overnight at 4°C and subsequently incubated with fluorescently labeled secondary antibodies (1:500) for 1 hr at room temperature. Primary antibodies: Hypoxyprobe-1 Plus (Hypoxyprobe, 4.3.11.3); PCNA (Abcam, ab29); CD31 (Abcam, ab182981). For hypoxyprobe primary signal was amplified with either Alexa Fluor secondary antibodies (Invitrogen) or a tyramide signal amplification kit (Invitrogen, T20924) per manufacturer's instructions. Administration of Hypoxyprobe-1 was as previously described (*Sammarco et al., 2014*). For immunofluorescence quantification, ImageJ was used to quantify images. Image subtraction was used to isolate fluorescence within the DAPI$^+$ nuclei. For Hypoxyprobe signal, the fluorescent intensity of Hypoxyprobe was measured, adjusted for background signal outside of the digit. For PCNA quantification, PCNA fluorescence was thresholded and normalized to the number of positive nuclei.

## Cell culture and metabolic assays

For Seahorse analysis with blastema, a single blastema was dissected from the digit at day 10, removing all nail, skin, bone, and soft tissue and placed directly into an Agilent Seahorse culture plate containing blastema media (*Sammarco et al., 2014*). Cells were allowed to expand in the well and were assayed after 2 weeks. Complete seahorse medium was prepared from Agilent Seahorse XF Base Medium (Agilent, 102353) to contain 10 mM glucose (Agilent, 103577-100), 2 mM glutamine (Agilent, 103579-100), and 1 mM pyruvate (Agilent, 103578-100), adjusted to pH 7.4. Cells were incubated in 180 µL complete seahorse medium at 37°C for 1 hr before measurements in a Seahorse XFe24 analyzer. Stressor concentrations were as follows: 2 µM oligomycin, 0.75 µM FCCP, 1 µM rotenone, and 1 µM antimycin A. Cells were evaluated using the Seahorse Bioscience Mito Stress Test (Agilent, Santa Clara, CA) using the standard protocol after optimization of both cell seeding density and FCCP concentration as previously reported (*Busse et al., 2019*). OCR and ECAR were normalized to total protein. For the Glycolytic Rate Assay, complete seahorse medium was prepared from Agilent Seahorse XF Base Medium to contain 11 mM glucose, 1 mM sodium pyruvate (Gibco, A24940-01), 2 mM glutamine (Agilent, 103579-100), and 5 mM HEPES (Gibco, 103337-100) adjusted to pH 7.4. Blastemas were treated with 2 mM OAA for 24 hr prior to Seahorse analysis (Sigma, 07753).

## Micro-computed tomography

Digits were scanned on a Bruker SkyScan 1172 at a pixel size of 4 µm with 0.2 rotation angle and 5 frame averaging using a 0.25 mm aluminum filter. The X-ray source used was 50 kV, 201 µA, and 10 W. The raw scans were reconstructed using NRecon. Each scan was reconstructed with a beam hardening correction of 24%, no smoothing correction, and a dynamic range of 0.00–0.339. Reconstructed output files were in 8-bit BMP format. Digits were rotated three dimensionally in DataViewer during which the regenerated woven bone was separated from the cortical stump. Whole bone and trabecular bone data sets were binarized in CTAn. Global thresholds were used for all data sets with minimum threshold value of 0 and maximum threshold value of 74. P3 was then cropped away from P2 in three dimensions using the 3D Viewer plugin in ImageJ2. Bone volume for whole bone and trabecular bone was generated using the volume function in the BoneJ plugin for ImageJ2. Trabecular and cortical

thickness values were generated using the thickness function in the BoneJ plugin. Trabecular spacing stacks were generated from the trabecular bone data sets using the background subtract function in ImageJ2. A rolling ball value of 29 was chosen such that the marrow cavity space was excluded but all trabecular spaces were included during the background subtract function. The despeckle function was then run to remove artificially small spaces on the exterior of the bone. Trabecular spacing values were generated using the thickness function in BoneJ. For 2D analysis, scans were realigned along the proximal-distal axis and an ROI was selected, beginning at the most proximal slice free of bone from the residual amputated stump. Images were thresholded as above and 2D measurements of bone area were generated in CTan (Bruker). Inner void fraction and 3D translucent images were quantified as described previously (*Hoffseth et al., 2021*). For OAA-treated micro-CT analysis, digits were aligned so that image stacks were aligned along the proximal-distal axis. The regenerated region was isolated based on the increased porosity and decreased cortical bone thickness associated with the transition from residual bone stump to newly deposited bone. 2D and 3D analyses for these digits were conducted in CTan.

## Spatial transcriptomics

Spatial transcriptomics was conducted using the Visium Spatial Gene Expression System (10× Genomics, Pleasanton, CA). Digits 10 days after amputation were harvested, arrayed in a 2 × 2 grid, fresh frozen in OCT, then stored at –80°C. Samples were cut at –26°C at a thickness of 12 µm. Optimization and gene expression assays were carried out according to manufacturer's protocol. Briefly, slides were fixed in –20°C methanol, dried with isopropanol, and stained with H&E. A tile scan image of all reaction areas was generated using a Leica DM6 B microscope (Leica Microsystems Inc). For tissue optimization, enzymatic permeabilization was conducted for 0–40 min, followed by first-strand cDNA synthesis with fluorescent nucleotides. The slide was reimaged using standard Texas Red filter cube. An optimal permeabilization time of 13 min was determined by visual inspection to maximize mRNA recovery while at the same time minimizing diffusion. For gene expression, initial workflow was consistent with optimization. Library preparation, clean-up, and indexing were conducted using standard procedures. Samples were subjected to pair-ended sequencing using an Illumina HiSeq generating ~400 M reads. Alignment and demultiplexing were conducted using the *SpaceRanger* pipeline with subsequent analyses conducted using *Seurat* in *R*. Pathway analysis was conducted using DAVID. Pathway scoring was determined using the *AddModuleScore* function in Seurat using gene lists obtained from validated KEGG pathways and published literature. Spatial feature plots represent a representative image, while all quantitative measures represent the cumulative data of multiple digits (5 young digits, 7 aged digits; 4 aged controls, 7 OAA-treated aged). To determine significance between genes/modules in spatial, the Wilcox method of *stat_compare_means* function from the *R* package *ggpubr* was used. For SpatialTime analysis, a reference line was manually drawn along the residual bone stump. Minimum 2D geometric distances were calculated between each spatial spot within the defined blastema and this reference line using the spatial coordinates and individually scaled to values between 0 (spatial spot closes to the bone stump) and 1 (furthest spot from the bone stump) for each blastema. Once the SpatialTime values were calculated for each blastema, samples are concatenated to increase statistical power and visualize gene expression across an 'average' blastema.

## ScRNAseq and 'blastema gene signature'

Previously published scRNAseq data sets were downloaded from the Gene Expression Omnibus repository and analyzed using default settings in Seurat to replicate original publication as closely as possible. For data from Storer et al., day 10 and 14 data sets (GSE135985) were used. For data from Johnson et al., day 11 data (GSE143888) was used. For the blastema signatures from our spatial data, blastema-specific DEGs (*Supplementary file 1*), the top 200 genes were highlighted after sorting for first the adjusted p-value, then the log fold change, and finally for specificity (defined as the ratio between spatial spots which expressed each gene within the blastema relative to spots expressing the gene outside the blastema). Genes highlighted in each of the three selection criteria were chosen first, with subsequent genes highlighted in two of the three criteria (weighted with specificity >log fold change >adjusted p-value) later selected to reach a list of 100 genes. Similar selection criteria were applied to Table S1A from Johnson et al. for genes enriched in fibroblast clusters. Following initial

selection, genes derived from fibroblast cluster 8 from Johnson et al. were excluded to generate a final blastema gene list. To generate a blastema signature from Storer et al., the top 100 gene entries were selected from category 1 genes in Table S2 from the original publication. As described above, module scoring was used to show enrichment of blastema gene signatures in each of the individual scRNAseq data sets.

### Statistical analysis

Statistical analysis was performed using SPSS (v.26, IBM) and Graphs were compiled using GraphPad Prism (version 7) (GraphPad, San Diego, CA).

Micro-CT data: Eight digits were precluded from the analysis due to breakage, as determined by visual inspection of 3D digit renderings. Unless otherwise stated, outliers were removed where an outlier was defined as having a studentized residual >3.0 or <–3.0, or an unstandardized residual greater or less than three times the group interquartile range. Normality of each group was tested using the Shapiro-Wilk test. Homogeneity of variances was tested using Levene's test of equality of variances. For all tests, $p \leq 0.05$ was considered significant. The data are presented as the means $\pm$ SD. For unamputated digit volume, a one-way ANOVA with Welch's test was performed, N=11 (6 months) and 14 (18 months). Differences in day 0-normalized P3 bone volume were determined by two-way ANOVA, N=63 (18 months) and 86 (6 months). Following the omnibus two-way ANOVA, individual differences between age groups at each time point were determined by post hoc comparisons using Bonferroni's adjustment for multiple comparisons.

Seahorse data: The statistical significance of the differences in the means of the investigated cell lines was determined by unpaired Student's t test. For all tests, $p \leq 0.05$ was considered significant. The data are presented as the means $\pm$ SD.

Hypoxyprobe and IHC: Age-associated differences were assessed using a one-way random effects ANOVA to account for repeated measures at the mouse level. Measurements were log-transformed to reduce skewness and results are reported on the logarithmic scale.

## Acknowledgements

This work was supported by a research grant from the National Institute of General Medical Sciences P20GM103629 (Sammarco).

## Additional information

### Funding

| Funder | Grant reference number | Author |
|---|---|---|
| National Institute of General Medical Sciences | P20GM103629 | Mimi C Sammarco |

The funders had no role in study design, data collection and interpretation, or the decision to submit the work for publication.

### Author contributions

Robert J Tower, Conceptualization, Data curation, Formal analysis, Investigation, Methodology, Software, Validation, Writing – original draft, Writing – review and editing; Emily Busse, Data curation, Formal analysis, Investigation, Methodology, Validation, Writing – review and editing; Josue Jaramillo, Data curation, Formal analysis, Investigation, Visualization, Writing – review and editing; Michelle Lacey, Data curation, Formal analysis, Writing – review and editing; Kevin Hoffseth, Formal analysis, Methodology, Writing – review and editing; Anyonya R Guntur, Conceptualization, Formal analysis, Methodology, Writing – review and editing; Jennifer Simkin, Conceptualization, Formal analysis, Supervision, Writing – original draft, Writing – review and editing; Mimi C Sammarco, Conceptualization, Data curation, Formal analysis, Funding acquisition, Investigation, Methodology, Project administration, Resources, Supervision, Validation, Visualization, Writing – original draft, Writing – review and editing

## Author ORCIDs
Robert J Tower http://orcid.org/0000-0001-5856-5758
Mimi C Sammarco http://orcid.org/0000-0001-9491-5735

## Ethics
All experiments were performed in accordance with the standard operating procedures approved by the Institutional Animal Care and Use Committee of Tulane University Health Sciences Center.

## Decision letter and Author response
Decision letter https://doi.org/10.7554/eLife.71542.sa1
Author response https://doi.org/10.7554/eLife.71542.sa2

---

# Additional files

## Supplementary files
• Supplementary file 1. Differentially expressed genes (DEGs) preferentially expressed in the blastema.

• Supplementary file 2. Differentially expressed genes (DEGs) preferentially expressed in the boundary.

• Supplementary file 3. Differentially expressed genes (DEGs) preferentially expressed in the remaining digit.

• Supplementary file 4. Blastema fingerprint genes identified from published single-cell RNA sequencing (scRNAseq) and spatial transcriptomics.

• Supplementary file 5. Blastema differentially expressed genes (DEGs) differentially expressed between young and aged mice.

• Supplementary file 6. Spatial differentially expressed genes (DEGs) differentially expressed between control and oxaloacetate (OAA)-treated mice.

• Transparent reporting form

## Data availability
Spatial transcriptomic data from day 10 young and aged mice, as well as day 21 aged mice treated with OAA or saline control have been deposited in the Gene Expression Omnibus (GEO) under the accession number GSE180682.

The following dataset was generated:

The following dataset was generated:

| Author(s) | Year | Dataset title | Dataset URL | Database and Identifier |
|---|---|---|---|---|
| Sammarco MC, Tower RJ | 2022 | Spatial transcriptomics reveals increased energetic requirements underpinning age-dependent declines in digit regeneration rescued through administration of OAA | https://www.ncbi.nlm.nih.gov/geo/query/acc.cgi?&acc=GSE180682 | NCBI Gene Expression Omnibus, GSE180682 |

The following previously published datasets were used:

| Author(s) | Year | Dataset title | Dataset URL | Database and Identifier |
|---|---|---|---|---|
| Storer MA | 2019 | Acquisition of a unique mesenchymal precursor-like blastema state underlies successful adult mammalian digit tip regeneration | https://www.ncbi.nlm.nih.gov/geo/query/acc.cgi?acc=GSE135985 | NCBI Gene Expression Omnibus, GSE135985 |
| Lehoczky JA | 2020 | Single cell RNA sequencing of adult regenerating mouse digit tips | https://www.ncbi.nlm.nih.gov/geo/query/acc.cgi?acc=GSE143888 | NCBI Gene Expression Omnibus, GSE143888 |

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
