## [Editor Report]

This paper will be of interest to a broad range of scientists in the regeneration field as it builds on and complements several recent mouse digit tip regeneration single-cell RNAseq data sets. This study applies the emerging field of spatial transcriptomics to overlay gene expression information on the spatial arrangement of regenerating cells over time. The authors use their data to address several important questions related to regeneration such as "can we define a molecular signature for regenerating cells?" and "why does regenerative ability decline with age?

---

## [Decision Letter]

**Decision letter after peer review:**

Thank you for submitting your article "Spatial transcriptomics reveals increased energetic requirements underpinning age-dependent declines in digit regeneration rescued through administration of OAA" for consideration by *eLife*. Your article has been reviewed by 3 peer reviewers, and the evaluation has been overseen by a Reviewing Editor and Kathryn Cheah as the Senior Editor. The reviewers have opted to remain anonymous.

Essential revisions:

1. The introduction is very abbreviated, uses a large number of field-specific terms, and several places could be expanded. For example, the description of different phases of digit regeneration should be expanded as this is critical information to guide non-digit regeneration aficionados through the rest of the manuscript's results. Likewise, for a large portion of the text, information is contained in parentheses when it should just be expanded into its own sentence. This would help with the flow of information in the manuscript.

2. There is no mention in the text or methods as to the X/Y/Z resolution that is achieved via spatial transcriptomics. Some approximation should be made to the number of cells per barcoded spot.

3. The PCNA staining between young and aged digits should be quantified with consideration of total cell number and density. It is not overwhelmingly obvious that there is a difference.

To distinguish whether there is in fact increased percentage of proliferating cells and/or a lengthening of the aged digit cell cycle, the authors should consider a BrdU/EdU pulse-chase that can give kinetics of the cell cycle stages between young and aged regenerating digits.

4. Despite the impressive analysis presented, the manuscript would benefit from a discussion of the limitations of the spatial transcriptomics technique as the low read depth in single cell transcriptomes are known to limit the detection of important genes that have low expression values, and this could be further compounded by the slide-based barcoding technique. An example of such a gene would be PTPRC where the authors show in Figure 1A a lack of spatial detection of a gene carried by all immune cells. In conflict with this, the authors previous work had already identified a large number of PTPRC+ cells within the mouse digit at this time point (Development. 2017 Nov 1; 144(21): 3907-3916. doi: 10.1242/dev.150086).

5. As aged digits are slower to enter the anabolic phase of bone growth it is possible that the "spatial time" data obtained at a single timepoint may simply reflect that delay. The authors should acknowledge this limitation in interpretation. If there would be a delay in the progression of blastema and bone formation between young and aged mice, as shown in Figure 2A, would experimental samples at the same days post amputation be equivalent samples between aged and young digits? Perhaps the authors have done this analysis of heterochronic timepoint comparison, but this is not immediately apparent in the methods or text. The reviewers had a difficult time understanding how comparisons were made between young and aged cohorts.

6. One of the primary differences uncovered by the authors is the increase in the number of cells expressing the active cell cycle marker PCNA (although this is not quantified) as well as an increase in expression of S/G2/M phase genes in aged digits relative to young regenerating digits. The interpretation by the authors is that there is increased proliferation in young versus aged regenerating digits. This was interpreted to mean that a greater percentage of all cells are actively moving through the cell cycle as opposed to remaining quiescent. An alternative explanation is that there is a lengthening of the S/G2/M phases in aged cells resulting in a higher proportion of cells at a given static timepoint being in these phases. These two interpretations are not mutually exclusive, but the authors do not consider a lengthening of the cell cycle and the impact that might have on the overall process of regeneration. A slower cell cycle could potentially explain lags in blastema/bone formation. The conclusions regarding cell cycle therefore must be tempered and other possible conclusions acknowledged.

7. The authors postulate in the discussion that there could be an effect on cell dedifferentiation and redifferentiation that correlates to the lag in bone anabolism. "Most studies agree that formation of the blastema relies, at least in part, on the dedifferentiation of terminal cells within the remaining digit (6, 7, 28, 29). In contrast, proliferation is thought to play a relatively less critical contribution to blastema formation (6,7)." Can the authors easily support this claim and perhaps demonstrate deficiencies in aged digits by ratiometric comparison of spatial expression of genes that would relate to dedifferentiation (or lack of differentiation) relative to proliferation and cell cycle markers?

8. The reviewers wondered if authors had considered that the observed hypoxia might be influencing osteoblast differentiation, which is leading to a lag in bone anabolism? There is literature suggesting that hypoxia can influence differentiation of osteogenic cells, although studies seem mixed as to whether hypoxia is beneficial or suppresses differentiation (one example is PubMed ID: 16529738). This seems like a question that the authors can easily address using their spatial datasets and immunohistochemistry in addition to their already generated in vitro assays.

9. The authors claim that the increased energetic demand drives intracellular hypoxia and increased blood vessel formation providing both spatial transcriptomic data, "hypoxyprobe" data and blood vessel immunofluorescent staining to support these claims. This data would be more convincing and reliable if the relative levels were quantified between old and young digits and if it was made clear where the vascular changes occurred relative to the spatial transcriptomics.

10. Figure 3B: Reviewers were confused about which regions of the spatial transcriptomics profile went into the identification of differentially expressed genes between young and aged mice. In Figure 1B, the authors have highlighted a specific portion of the image as the blastemal (pink color). In the text describing Figure 3, the authors state, "…Pathway analysis of DEGs showing differential expression in the young or old blastemal identified…(Figure 3A) " and then later state "…Metabolic analysis showed increased glycolysis and oxidative phosphorylation (OxPhos) in the blastema of aged mice… (Figure 3B)". It is confusing, therefore, that the images shown in Panel 3B show expression patterns in regions of the tissue that are not necessarily localized to the blastema. For example, OxPhos signatures for the young mice seem to be concentrated below the blastema in the digit itself; this raises the question of whether the authors segmented their gene lists to the blastema only while analyzing pathways, or whether the whole section was considered? Perhaps the authors could outline the region of the blastema with a dashed line, to emphasize where their pathway analyses were focused? If otherwise, please clarify. Notably, a similar concern could be raised about Figure 4B and 4C and should be likewise addressed in these areas.

11. The concern was raised that the data showing in Figure 4A to be somewhat unnecessary for the paper. While the Figure 3 cell energetics data are quite relevant as they are from young and aged blastema, the data in Figure 4A are only provided from an osteogenic cell line (which has limited relevance to this study overall). Presumably these data are provided to link the exciting cell energetics data in Figure 3 to the hypoxia data in Figure 4. The reviewers recommended that the data be removed from the paper, moved to a supplementary figure as they detract from (rather than add to) the nice narrative the authors have otherwise put together. The reviewers requested that these experiments be repeated with blastema tissue rather than osteogenic cells, which would provide a much better linkage between the story told in Figure 3 and the continued story in Figure 4. A similar concern could be raised about Figure 5A-B as was raised above for Figure 4. The authors have clearly demonstrated the ability to perform these assays with blastema tissue (Figure 3D and S4C) and the justification for moving to a cell line is thin and unsupported. The reviewers were concerned because the relative importance of the Seahorse assays must be assumed to translate from osteogenic cells to blastema, rather than being directly supported with experimental evidence.

12. In Figure 3D, the authors state that OCR associated with ATP, maximal, and non-mitochondrial respiration were significantly higher in aged vs. young blastemas. However, these data are not shown anywhere, and statistical analyses are not presented in Panel D. Please add these graphs and the related statistical analyses.

All of the reviewers struggled with the presentation and interpretation of the results of the OAA study. Overall, there are so much confusion about what OAA is doing and what these data mean, that it became difficult for the reviewers to reconcile this OAA study with the hypotheses and premise of the rest of the paper. The following two comments address this part of the paper and are considered essential issues that must be resolved.

13. The choice of intervention (treatment with the anapleurotic metabolite oxaloacetate (OAA)) is hard to follow and poorly justified. The authors showed that the aged mice already had elevated OCR already vs. young mice (Figure 3B,C,D), as well as enhanced proliferation (Figure S4C). The data they provide in Figure 5 show that OAA increases OCR and proliferation. So, why is this drug a good choice when it produces the same phenotype already seen in the aged mice? This part of the story, accordingly, becomes hard to follow. As written, the rationale is unclear and should be clarified for the reader. This is particularly important because OAA did not rescue the hypoxia phenotype seen in the aged mice, despite providing a modest rescue of bone regeneration endpoints. While it is recognized that the authors likely used the MC3T3 osteoblast cell line to restrict analysis to bone producing cells to match the in vivo enhanced bone anabolism phenotype, performing the OAA treatments ex vivo, in primary culture of aged blastema tissue using procedures they have already documented (Figure 3) is a necessary addition for their conclusions to be supported.

14. The failure of OAA to alter trabecular separation and a full rescue of the aged phenotype data broadly support this hypothesis although a discussion of alternative hypotheses are not discussed (i.e., cell specific effects of OAA on cell phenotype, alternative metabolic pathway activation with excess OAA, mineralization specific gene activation etc..?). It appears that OAA acted so in a way that is apparently largely outside of the pathways identified in the spatial transcriptomics analyses. The documentation of timeline for any changes in the initiation of bone anabolism (outgrowth) upon OAA treatment are strangely missing and would be important for understanding if metabolic energy deficit is responsible for the delay or if some other pathway is involved. The actual mechanism of action of OAA is never defined but the addition of OAA seems to have decoupled bone anabolism from hypoxia, so this may not be the case. Perhaps there are non-osteogenic cells that are primarily affected by hypoxia and these cells are not assayed in vitro? Do the authors have the spatial resolution to distinguish osteogenic cell gene expression from other non-osteogenic cell types?

15. There is something counterintuitive and unexplained in the authors data related to the aged regenerative phenotype. They first observe an increased proliferation in aged blastemas and increased expression of metabolic pathways. Postulating that cells are compensating for a metabolic deficit, they add an OxPhos metabolite OAA. in vitro, this metabolite increases proliferation and oxidative phosphorylation. It is not well explained what is being achieved through the rescue experiment relative to the baseline aged phenotype other than an increased mineralization. Is it because these assays were done with the MC3T3 cell line and not primary digit explants? This is not explained sufficiently, and the in vitro assays using cell lines may not be an adequate proxy for aged tissue cells.

Please address any comments and issues raised in addition to the essential reviewer comments listed above.

*Reviewer #1:*

The authors set out to apply the new technology of spatial transcriptomics to the context of digit tip regeneration and identify possible targets regulating the age-related decline in regeneration quality. The application of spatial transcriptomics to this model was extremely convincing as they were able to incorporate several published datasets into their pipeline. During their analysis they were able to spatially identify a blastemal region and a border region with different gene signatures to the rest of the digit.

The authors then confirm the morphological changes in aged vs young mice using rigorous bone measurements reporting the main defects in aged mice occurring which is in agreement with a previous study by Brunauer et al., characterizing the bone volume changes in young vs aged mice. This current study adds trabecular thickness and trabecular separation measurements which can add to the understanding of regeneration quality in aged mice.

The manuscript then focuses on metabolism as a major instigator for disruption of regenerative quality. The spatial transcriptomics approach is used to reveal changes in metabolism location in the proximo-distal axis in aged digits. This analysis could be really important for understanding the dynamic spatial nature of how cells communicate within a blastema. Additional ex vivo seahorse data is supplied showing changes in metabolic parameters of aged digit cells that supports the hypothesis that changes in the aged digits metabolism may be important in age related decline in regeneration quality. The authors claim that the increased energetic demand drives intracellular hypoxia.

Next the authors perform in vitro analysis on a cultured osteoblast cell line treated with oxaloacetate (OAA) demonstrating changes in cell energy use parameters, proliferation and mineralization. They then perform studies examining OAA treatment in vivo and show data on aged digits 28 days post-amputation measuring increased bone volume and thickness. The authors also demonstrate that OAA fails to modulate hypoxia in the aged digits and put forward the hypothesis that hypoxia driven angiogenesis results in the failure of OAA to alter trabecular separation and a full rescue of the aged phenotype. Overall, the data broadly support this hypothesis, however the question of if the metabolic energy deficit is responsible for the delay or if some other pathway is involved remains unresolved.

The work presented in this manuscript is of high interest to regeneration and developmental biology disciplines. One key problem in these specialties is the difficulty in obtaining spatial information of gene expression for a large number of genes simultaneously. One can use modern in situ hybridization techniques such as RNAscope or HCR fluorescent probe (biased) analyses to identify the location of a particular gene on a tissue section however this is limited to a small number of genes at a time. The non-biased spatial transcriptomics approach has many advantages and will advance the field significantly. The authors do a good job at demonstrating this advanced technique at a single point in the regeneration timeline and add some interesting insights into the role of metabolism in aged-related decline in regeneration quality.

*Reviewer #2:*

Spatial transcriptomics is a fairly newly emerging field where the transcriptome from a section of tissue can be ascertained while also retaining information regarding those transcripts' spatial location in the tissue – preserving important relational information which can increase the utility of the resulting molecular data. While this technique is becoming more widely used in certain model systems (e.g., zebrafish) and tissues / organ systems (e.g., brain), it had not yet been described in the context of regenerating musculoskeletal tissue. A major strength of the manuscript, which does exactly that, is that the authors perform these analyses and then compare their results against two other recent datasets – collected using single cell RNA sequencing, another technique increasing in popularity – and compare / contrast against their own findings. Many similarities exist between the datasets, which highlights the likelihood that both spatial transcriptomics and single cell RNA-seq techniques are providing an accurate description of the molecular events taking place in digit regeneration, but notable differences exist that highlight the particular strength of spatial transcriptomics technology in this circumstance.

Another strength of the manuscript is that it does not simply stop at proving the utility of spatial transcriptomics for tracking the molecular signature of regenerating tissue – the authors go further to apply their new technology to a known instance of impaired regeneration (aging), and then use spatial transcriptomics to define the molecular events that lead to that expected outcome. This provides novel data to the literature, in that the authors have identified (using their technology) a molecular signature of impaired cellular bioenergetics, which was then pursued with other experiments (e.g., Seahorse, hypoxia endpoints, etc.) to test how and why aging tissue fails to regenerate like younger tissue.

A weakness of the manuscript is that the choice of intervention (treatment with the anapleurotic metabolite oxaloacetate (OAA)) as this drug induced, rather than blunted, the same metabolic effects seen in the aged animals that the authors hypothesized would be causative for their phenotype. This drug also produced a partial rescue of the impaired bone regeneration phenotype but did so in a way that is apparently largely outside of the pathways identified as differentially regulated between aged and young mice identified in the spatial transcriptomics analyses.

*Reviewer #3:*

In the manuscript by Tower et al., the authors address an important consideration related to regeneration which is the complexity and dynamicity of the process is somewhat lost in bulk and single cell transcriptomic approaches. Using a spatial transcriptomic approach, the authors can compartmentalize the regeneration process to the stump tissue, the regenerative blastema, and the boundary zone. To highlight the utility of this approach, the authors re-analyze two recent scRNAseq datasets of digit regeneration to identify consensus gene sets that predict spatial proximity within the blastema, in contrast to clustering by cell type which is the norm for scRNAseq datasets. This is well done and provides good justification for why high throughput spatial gene expression studies are necessary. The authors then shift focus to look at a primary biological question related to regeneration which is why regenerative ability declines with age. Using young and aged cohorts of mice, the authors find a decline in the speed of regeneration in aged mice, specifically in the creation of new bone and its patterning relative to young regenerating or uninjured digits. Transcriptomic analysis suggested that three potentially interconnected processes were changed – cell proliferation, hypoxia, and increased glycolytic and OxPhos cell metabolism – all increased in aged digits. The authors posit that a metabolic deficit could be underlying changes to these processes and use a metabolite that enhances oxidative metabolism to see if they can rescue aged digit regeneration. This treatment is able to somewhat rescue bone growth but cannot rescue the observed hypoxia and bone density which seems to be due to the increased vascularization that is a consequence of hypoxia.

[Editors' note: further revisions were suggested prior to acceptance, as described below.]

Thank you for resubmitting your work entitled "Spatial transcriptomics reveals increased energetic requirements underpinning age-dependent declines in digit regeneration rescued through administration of OAA" for further consideration by *eLife*. Your revised article has been evaluated by Kathryn Cheah (Senior Editor) and a Reviewing Editor.

The manuscript has been improved but there are some remaining issues that needs to be addressed, as outlined below:

In Figure 2, the Day O, 7, 10, 12, 14, 21, 28 42 MicroCT time course is used to show changes in age related delayed bone anabolism. The reviewer and editor found it unfortunate that a OAA treated vs vehicle control aged mouse time course was not supplied to demonstrate this enhanced anabolism. Measurement of bone volume, trabecular thickness, trabecular number, and trabecular separation, of regenerated control and OAA treated bone at a single Day 28 timepoint does not tell the scientific community where the OAA is acting between 10 and 28 DPA. If these MicroCT measurements can be added reasonably quickly and would certainly help with the interpretation of the OAA effects on partially rescuing the aged bone anabolic defect. For example, if the OAA only altered bone volume between D21- D28 or if the OAA was acting earlier to advance the start the anabolism (D12-D14). If these measures cannot be collected quickly as the samples are not in hand, please expand the discussion to address this limitation of the paper. This will need to be done quickly to ensure the timely publication of this paper.

*Reviewer #1:*

The authors have made significant improvements to the clarity and presentation of the paper. They have addressed most of the comments by adding additional valuable data.

Although documentation of the timeline for any changes in the initiation of bone anabolism (outgrowth) upon OAA treatment are still strangely missing which would have been very informative for understanding if metabolic energy deficit is responsible for the delay or if some other pathway is involved. Despite this, the additional day 28 spatial transcriptomics data is a valuable addition to the manuscript that adds a deeper demonstration of the utility of this technique.

Overall, this is now a clear and informative manuscript of high value to the community.

[Editors' note: further revisions were suggested prior to acceptance, as described below.]

Thank you for resubmitting your work entitled "Spatial transcriptomics reveals increased energetic requirements underpinning age-dependent declines in digit regeneration rescued through administration of OAA" for further consideration by *eLife*. Your revised article has been evaluated by Kathryn Cheah (Senior Editor) and a Reviewing Editor.

The manuscript has been improved but there is one remaining issue that must be addressed, as outlined below:

The Age-related defect in P3 digit tip regeneration is most apparent in the timing of anabolism onset occurring between D12-D14 post-amputation. At D21 this manifests as a significant difference in Bone Volume. The Tb. thickness and Tb. spacing differences are not significant at D21 (in figure 2) and are only significant a D28. Taking this into consideration, the OAA treatment data presented (in figure 6) show no differences in bone volume, Tb. thickness and Tb. spacing at D21 and OAA effects are only really observed in volume and thickness at D28 with no differences in spacing. This could be interpreted as the effect of OAA not affecting the onset of anabolism at all, but rather providing support for late-stage remodeling phase. The authors MUST discuss this, as this interpretation contradicts the statement given by the authors that "This data fortifies our findings in spatial transcriptomics (D21) that indeed OAA acts in advance to the start of anabolism."

---

## [Author Response]

Essential revisions:1. The introduction is very abbreviated, uses a large number of field-specific terms, and several places could be expanded. For example, the description of different phases of digit regeneration should be expanded as this is critical information to guide non-digit regeneration aficionados through the rest of the manuscript's results. Likewise, for a large portion of the text, information is contained in parentheses when it should just be expanded into its own sentence. This would help with the flow of information in the manuscript.

We appreciate this recommendation. We have expanded the introduction to address the different phases of digit regeneration, included information in parentheses, and included explanations of field-specific terms. We hope this improves the flow of information.

2. There is no mention in the text or methods as to the X/Y/Z resolution that is achieved via spatial transcriptomics. Some approximation should be made to the number of cells per barcoded spot.

Additional text has been added to clarify the spatial resolution of this technique (55 µm), as well the thickness of the section used for library construction (12 µm). To calculate the approximate number of cells present within each spatial spot, DAPI stained blastema at D10 following amputation were segmented into representative 55um circular ROIs and the number of nuclei determined. These estimations suggest each spatial spot likely corresponds to ~5-10 cells depending on the region of the blastema analyzed. This information has now been included in the manuscript.

3. The PCNA staining between young and aged digits should be quantified with consideration of total cell number and density. It is not overwhelmingly obvious that there is a difference.To distinguish whether there is in fact increased percentage of proliferating cells and/or a lengthening of the aged digit cell cycle, the authors should consider a BrdU/EdU pulse-chase that can give kinetics of the cell cycle stages between young and aged regenerating digits.

We appreciate this recommendation and have quantified the PCNA staining (Figure 3D) with consideration of total cell number. This number represents the ratio of PCNA positive nuclei ± SD in aged blastemas relative to young, with values over 1 indicating increased proliferation in aged blastema. Additionally, we have included spatial transcriptomic analysis of genes linked to cell cycle activity, including cyclins and components of the anaphase promoting complex (Figure 3E) to further support that genes linked to entry into, and progression through, the cell cycle is elevated in the aged blastema.

4. Despite the impressive analysis presented, the manuscript would benefit from a discussion of the limitations of the spatial transcriptomics technique as the low read depth in single cell transcriptomes are known to limit the detection of important genes that have low expression values, and this could be further compounded by the slide-based barcoding technique. An example of such a gene would be PTPRC where the authors show in Figure 1A a lack of spatial detection of a gene carried by all immune cells. In conflict with this, the authors previous work had already identified a large number of PTPRC+ cells within the mouse digit at this time point (Development. 2017 Nov 1; 144(21): 3907-3916. doi: 10.1242/dev.150086).

Additional text has now been added to the discussion to clearly state the limitations associated with the low efficiency conversion of the transcriptome into cDNA for sequencing. This limitation has led us to primarily focus on overall comparisons between blastemas, the use of module scoring (which relies on a list of genes rather than an individual marker) and the use of SpatialTime which allows the concatenation of samples to display an ‘average’ blastema across space with increased statistical power.

5. As aged digits are slower to enter the anabolic phase of bone growth it is possible that the "spatial time" data obtained at a single timepoint may simply reflect that delay. The authors should acknowledge this limitation in interpretation. If there would be a delay in the progression of blastema and bone formation between young and aged mice, as shown in Figure 2A, would experimental samples at the same days post amputation be equivalent samples between aged and young digits? Perhaps the authors have done this analysis of heterochronic timepoint comparison, but this is not immediately apparent in the methods or text. The reviewers had a difficult time understanding how comparisons were made between young and aged cohorts.

To address differences in gene expression we analyze both the aged and young blastema 10 days after amputation. As stated by the reviewer, this approach may capture differences that are merely representative of a similar, but delayed, gene expression pattern in the aged sample relative to young controls. However, our studies in hypoxia and proliferation suggest that these gene expression differences are not simply a delay, but an actual shift in cell function that sustains throughout regeneration. We have added discussion of this limitation to the text of the manuscript.

6. One of the primary differences uncovered by the authors is the increase in the number of cells expressing the active cell cycle marker PCNA (although this is not quantified) as well as an increase in expression of S/G2/M phase genes in aged digits relative to young regenerating digits. The interpretation by the authors is that there is increased proliferation in young versus aged regenerating digits. This was interpreted to mean that a greater percentage of all cells are actively moving through the cell cycle as opposed to remaining quiescent. An alternative explanation is that there is a lengthening of the S/G2/M phases in aged cells resulting in a higher proportion of cells at a given static timepoint being in these phases. These two interpretations are not mutually exclusive, but the authors do not consider a lengthening of the cell cycle and the impact that might have on the overall process of regeneration. A slower cell cycle could potentially explain lags in blastema/bone formation. The conclusions regarding cell cycle therefore must be tempered and other possible conclusions acknowledged.

To address this concern, we looked at the expression of genes linked to cell cycle progression and cyclin destruction (Figure 3E). These new data suggests not only high levels of cyclin transcripts, but also several genes linked to cyclin destruction and cell cycle progression. While analysis of transcripts does not provide definitive evidence to distinguish these two possibilities, these new data supports active progression through the cell cycle over lengthening or delays of progression. Text has been added to the results to reflect these new data, as well as in the discussion to temper our original conclusion.

7. The authors postulate in the discussion that there could be an effect on cell dedifferentiation and redifferentiation that correlates to the lag in bone anabolism. "Most studies agree that formation of the blastema relies, at least in part, on the dedifferentiation of terminal cells within the remaining digit (6, 7, 28, 29). In contrast, proliferation is thought to play a relatively less critical contribution to blastema formation (6,7)." Can the authors easily support this claim and perhaps demonstrate deficiencies in aged digits by ratiometric comparison of spatial expression of genes that would relate to dedifferentiation (or lack of differentiation) relative to proliferation and cell cycle markers?

We agree with this comment that showing reduced dedifferentiation in aging mice would be ideally suited in our model. However, we were unable to find a clear transcriptional based approach to quantify dedifferentiation in young or aged blastemas. Many pathways, such as Notch and Pi3K-Akt pathways have been linked to dedifferentiation, but also in proliferation and differentiation. As such, we feel presenting these results as dedifferentiation markers may be misleading. Further, many additional markers previously proposed were not found to be expressed at meaningful levels within the blastema, likely owing to their low expression and the relatively low coverage of the whole transcriptome provided by spatial (this limitation is now stated in the discussion). In addition, we probed several genes linked to osteogenic differentiation. Seen in Author response image 1, our results show that while several genes were significantly downregulated (Runx2, Spp1), no strong evidence was obtained that osteogenesis was impaired in aged mice. This may be the result of the fact that our samples were obtained at day 10, while mineralization typically does not initiate until approximately day 14.

**Author response image 1. sa2fig1:** No notable shift in osteogenic transcript expression is observed by spatial transcriptomics. Relative expression of osteogenic genes in the blastema of young and aged mice 10 days post amputation. This osteoscore is comprised of the following genes: *Alpl, Bglap, Bglap2, Col1a1, Col1a2, Dmp1, Ibsp, Mef2c, Postn, Runx2, Sp7, Sparc, Phex, Satb2, Pth1r, Ostn, Car3*.

8. The reviewers wondered if authors had considered that the observed hypoxia might be influencing osteoblast differentiation, which is leading to a lag in bone anabolism? There is literature suggesting that hypoxia can influence differentiation of osteogenic cells, although studies seem mixed as to whether hypoxia is beneficial or suppresses differentiation (one example is PubMed ID: 16529738). This seems like a question that the authors can easily address using their spatial datasets and immunohistochemistry in addition to their already generated in vitro assays.

The hypoxia data from the Arnett Lab indeed provides an excellent foundation for what is the relationship between hypoxia and osteoblasts, and provided the motivation for some of our earlier studies in oxygen. Interestingly, we found that hypoxia delayed bone mineralization in the blastema (PMID: 24753124) during digit regeneration and noted this deviation from the Arnett lab findings. While spatial would be an excellent approach to further study this relationship, particularly in the blastema, our new data showing that OAA increases bone volume and thickness without decreasing hypoxia (comparably to the untreated mice) certainly adds a layer of complexity to the relationship between hypoxia and regeneration, at least in the digit model. Additionally, Figure R1 (above, #7 response) shows no strong evidence that osteogenesis was impaired in aged mice. As with dedifferentiation, this may be a result of the fact that our samples were obtained at day 10, prior to bone formation. We find our data intriguing and hope to further explore this relationship with multiple timepoints across regeneration in the future with more timepoints.

9. The authors claim that the increased energetic demand drives intracellular hypoxia and increased blood vessel formation providing both spatial transcriptomic data, "hypoxyprobe" data and blood vessel immunofluorescent staining to support these claims. This data would be more convincing and reliable if the relative levels were quantified between old and young digits and if it was made clear where the vascular changes occurred relative to the spatial transcriptomics.

We apologize for the lack of clarity when describing the data. Numbers below the spatial (Figure 5A) and hypoxyprobe images (Figure 5B) denote quantification of gene expression and average fluorescent intensity, respectively, within the blastema of aged mice, normalized to young (values greater than 0 indicating higher levels of transcripts/fluorescence in aged blastemas). Text has been added to clarify this quantification. We have also added additional analysis to show the expression level of our vessel scoring relative to SpatialTime (Figure 5E) to show that not only is vascularization increased in the aged blastema, but that these vessels are present throughout the blastema.

10. Figure 3B: Reviewers were confused about which regions of the spatial transcriptomics profile went into the identification of differentially expressed genes between young and aged mice. In Figure 1B, the authors have highlighted a specific portion of the image as the blastemal (pink color). In the text describing Figure 3, the authors state, "…Pathway analysis of DEGs showing differential expression in the young or old blastemal identified…(Figure 3A) " and then later state "…Metabolic analysis showed increased glycolysis and oxidative phosphorylation (OxPhos) in the blastema of aged mice… (Figure 3B)". It is confusing, therefore, that the images shown in Panel 3B show expression patterns in regions of the tissue that are not necessarily localized to the blastema. For example, OxPhos signatures for the young mice seem to be concentrated below the blastema in the digit itself; this raises the question of whether the authors segmented their gene lists to the blastema only while analyzing pathways, or whether the whole section was considered? Perhaps the authors could outline the region of the blastema with a dashed line, to emphasize where their pathway analyses were focused? If otherwise, please clarify. Notably, a similar concern could be raised about Figure 4B and 4C and should be likewise addressed in these areas.

We apologize for the confusion. All analyses, unless otherwise stated, were confined to the blastema as defined by Figure 1B. To better demonstrate this point, figures 3B, 4, 5, 7, S4 and S5 have been updated to reflect signal within the blastema only. Hopefully the reviewers will find these new representative images more precise and easier to interpret with regard to the corresponding quantitation.

11. The concern was raised that the data showing in Figure 4A to be somewhat unnecessary for the paper. While the Figure 3 cell energetics data are quite relevant as they are from young and aged blastema, the data in Figure 4A are only provided from an osteogenic cell line (which has limited relevance to this study overall). Presumably these data are provided to link the exciting cell energetics data in Figure 3 to the hypoxia data in Figure 4. The reviewers recommended that the data be removed from the paper, moved to a supplementary figure as they detract from (rather than add to) the nice narrative the authors have otherwise put together. The reviewers requested that these experiments be repeated with blastema tissue rather than osteogenic cells, which would provide a much better linkage between the story told in Figure 3 and the continued story in Figure 4. A similar concern could be raised about Figure 5A-B as was raised above for Figure 4. The authors have clearly demonstrated the ability to perform these assays with blastema tissue (Figure 3D and S4C) and the justification for moving to a cell line is thin and unsupported. The reviewers were concerned because the relative importance of the Seahorse assays must be assumed to translate from osteogenic cells to blastema, rather than being directly supported with experimental evidence.

Consistent with the reviewer’s request, we have removed the MC3T3 cell and now present Seahorse data from blastema cells treated with OAA (Figure 6A). In addition, to further characterize the effects of OAA in vivo, new spatial transcriptomic experiments were conducted (Figure 7) to better unravel the mechanism of action through which OAA rescues our impaired regeneration phenotype.

12. In Figure 3D, the authors state that OCR associated with ATP, maximal, and non-mitochondrial respiration were significantly higher in aged vs. young blastemas. However, these data are not shown anywhere, and statistical analyses are not presented in Panel D. Please add these graphs and the related statistical analyses.

We apologize for not including the statistical analyses relevant to the kinetic data. We have edited figures 4 and 6 to indicate specific time-point analysis and statistical significance for the third read after start and the third read after each stressor injection as our team has previously shown for OCR associated with basal, ATP, maximal, and non-mitochondrial respiration (PMID: 30953371).

All of the reviewers struggled with the presentation and interpretation of the results of the OAA study. Overall, there are so much confusion about what OAA is doing and what these data mean, that it became difficult for the reviewers to reconcile this OAA study with the hypotheses and premise of the rest of the paper. The following two comments address this part of the paper and are considered essential issues that must be resolved.13. The choice of intervention (treatment with the anapleurotic metabolite oxaloacetate (OAA)) is hard to follow and poorly justified. The authors showed that the aged mice already had elevated OCR already vs. young mice (Figure 3B,C,D), as well as enhanced proliferation (Figure S4C). The data they provide in Figure 5 show that OAA increases OCR and proliferation. So, why is this drug a good choice when it produces the same phenotype already seen in the aged mice? This part of the story, accordingly, becomes hard to follow. As written, the rationale is unclear and should be clarified for the reader. This is particularly important because OAA did not rescue the hypoxia phenotype seen in the aged mice, despite providing a modest rescue of bone regeneration endpoints. While it is recognized that the authors likely used the MC3T3 osteoblast cell line to restrict analysis to bone producing cells to match the in vivo enhanced bone anabolism phenotype, performing the OAA treatments ex vivo, in primary culture of aged blastema tissue using procedures they have already documented (Figure 3) is a necessary addition for their conclusions to be supported.

In the manuscript, we have better explained and justified our choice of OAA for the reviewers. The reviewer is correct that our data propose an increase in both glycolytic and oxidative metabolism pathways in the aged blastema as well as enhanced proliferation, suggesting an increased energy demand. Because of these data, we hypothesized that unmet energy needs may underpin age-dependent metabolic increases. Therefore, we tested if supplementing these cells with a substrate to off-set these increased energy requirements would enhance regeneration. OAA is able to directly generate NADH in the mitochondria. As such, we use this intervention as a way to test our hypothesis. We have modified the text that OAA rescue is likely mediated through helping to meet the new metabolic requirements of aged cells, rather than attempting to rescue and revert these aged cells back to a “young” phenotype. Additionally, we have added Seahorse analysis of OAA treated blastemas from aged mice. Results from this analysis show that administration of OAA reduced the maximal respiration of aged blastemas (D10) supporting a role of OAA in reducing energy demand by aged blastema cells.

14. The failure of OAA to alter trabecular separation and a full rescue of the aged phenotype data broadly support this hypothesis although a discussion of alternative hypotheses are not discussed (i.e., cell specific effects of OAA on cell phenotype, alternative metabolic pathway activation with excess OAA, mineralization specific gene activation etc..?). It appears that OAA acted so in a way that is apparently largely outside of the pathways identified in the spatial transcriptomics analyses. The documentation of timeline for any changes in the initiation of bone anabolism (outgrowth) upon OAA treatment are strangely missing and would be important for understanding if metabolic energy deficit is responsible for the delay or if some other pathway is involved. The actual mechanism of action of OAA is never defined but the addition of OAA seems to have decoupled bone anabolism from hypoxia, so this may not be the case. Perhaps there are non-osteogenic cells that are primarily affected by hypoxia and these cells are not assayed in vitro? Do the authors have the spatial resolution to distinguish osteogenic cell gene expression from other non-osteogenic cell types?

We appreciate this comment from the reviewers. To better understand the mode of action of OAA, we have included new data on OAA-treated mice using spatial transcriptomics. These data point to a divergent effect of OAA on differentiated and undifferentiated cells. Additional analyses indicate OAA promotes proliferation and suggest this rescue may be due in part through re-activation of the WNT pathway in aged digits. We have expanded the discussion to include OAA’s ability to only partially resolve the phenotype. While OAA was not able to resolve the intracellular hypoxia seen in aged mice, this outcome is consistent with our belief that OAA helps meet the energetic needs of highly metabolically active aged cells, rather than reversing metabolism back to a young-like state. We have additionally added language to describe the lack of rescue in trabecular spacing, likely due to the inability to resolve the VEGF signaling at day 21. It is interesting to note that OAA enhanced bone deposition with minimal changes in overall hypoxia signaling (Figure 7D,E), supporting the reviewer’s thoughts that OAA treatment seems to affect osteogenesis through mechanisms outside of hypoxia.

We have also placed greater focus on the idea that while signaling pathways such as WNT are known to modulate cell metabolism, here we show the converse where cell metabolism is altering signaling pathways (Figure 7), a mechanism that warrants further investigation in future studies. As a result, we have garnered new insight into the OAA-driven mechanisms that played a role in this. While the resolution does not allow us to distinguish which individual cell types express elevated transcripts linked to hypoxia, our new data support upregulation of WNT signaling as a potential mechanism through which OAA may rescue aging-impaired regeneration.

15. There is something counterintuitive and unexplained in the authors data related to the aged regenerative phenotype. They first observe an increased proliferation in aged blastemas and increased expression of metabolic pathways. Postulating that cells are compensating for a metabolic deficit, they add an OxPhos metabolite OAA. in vitro, this metabolite increases proliferation and oxidative phosphorylation. It is not well explained what is being achieved through the rescue experiment relative to the baseline aged phenotype other than an increased mineralization. Is it because these assays were done with the MC3T3 cell line and not primary digit explants? This is not explained sufficiently, and the in vitro assays using cell lines may not be an adequate proxy for aged tissue cells.

We appreciate this comment from the reviewers. To help explain, we have included language to support our choice of OAA, including its participation in the malate-aspartate shuttle and its ability to directly generate NADH in the mitochondria, hypothesizing that OAA would help meet the unmet bioenergetic need of the aged blastema. We have also included the data that drove this choice, showing that administration of OAA reduced the maximal respiration of aged blastemas (D10) (Figure 6A). Our additional spatial transcriptomics data analyzing newly formed bone in aged mice treated with OAA show that OAA in fact increases bioenergetics and proliferation in newly regenerated bone (D21) (Figure 7). We have built in language to better illustrate that this finding does not necessarily run counter to our objectives in this study. Our metabolic approach does not seek to reverse the aged phenotype so much as to provide the building blocks that are needed to fuel the increased bioenergetics in the aged blastema. We have also included language in the discussion highlighting the critical signaling of OxPhos-mediated ROS in limb regeneration and emphasizing the possibility that increased bioenergetics may be a critical piece of limb regeneration.

[Editors' note: further revisions were suggested prior to acceptance, as described below.]

The manuscript has been improved but there are some remaining issues that needs to be addressed, as outlined below:In Figure 2, the Day O, 7, 10, 12, 14, 21, 28 42 MicroCT time course is used to show changes in age related delayed bone anabolism. The reviewer and editor found it unfortunate that a OAA treated vs vehicle control aged mouse time course was not supplied to demonstrate this enhanced anabolism. Measurement of bone volume, trabecular thickness, trabecular number, and trabecular separation, of regenerated control and OAA treated bone at a single Day 28 timepoint does not tell the scientific community where the OAA is acting between 10 and 28 DPA. If these MicroCT measurements can be added reasonably quickly and would certainly help with the interpretation of the OAA effects on partially rescuing the aged bone anabolic defect. For example, if the OAA only altered bone volume between D21- D28 or if the OAA was acting earlier to advance the start the anabolism (D12-D14). If these measures cannot be collected quickly as the samples are not in hand, please expand the discussion to address this limitation of the paper. This will need to be done quickly to ensure the timely publication of this paper.

We appreciate this observation. OAA is administered to the mice between day 10 and 28. We have included microCT analysis for trabecular volume, thickness and spacing for day 21 and show that while volume and thickness appear to be increasing, this is not yet statistically significant relevant to the control group. This data fortifies our findings in spatial transcriptomics (D21) that indeed OAA acts in advance to the start of anabolism.

[Editors' note: further revisions were suggested prior to acceptance, as described below.]

The manuscript has been improved but there is one remaining issue that must be addressed, as outlined below:The Age-related defect in P3 digit tip regeneration is most apparent in the timing of anabolism onset occurring between D12-D14 post-amputation. At D21 this manifests as a significant difference in Bone Volume. The Tb. thickness and Tb. spacing differences are not significant at D21 (in figure 2) and are only significant a D28. Taking this into consideration, the OAA treatment data presented (in figure 6) show no differences in bone volume, Tb. thickness and Tb. spacing at D21 and OAA effects are only really observed in volume and thickness at D28 with no differences in spacing. This could be interpreted as the effect of OAA not affecting the onset of anabolism at all, but rather providing support for late-stage remodeling phase. The authors MUST discuss this, as this interpretation contradicts the statement given by the authors that "This data fortifies our findings in spatial transcriptomics (D21) that indeed OAA acts in advance to the start of anabolism."

We apologize for misunderstanding the requested revisions in our last submission. We are in complete agreement with the editors that we cannot completely discount the role of OAA in late-stage remodeling. Since the mouse was dosed with OAA through the blastema, early bone, and late bone stages, the OAA-dependent effect could come from OAA influence at any or all of these stages. Additional work is required to determine the exact mechanism through which OAA promotes regeneration. While we cannot rule out the role of OAA on promoting osteoblast mineralization during the late-stage remodeling phase, our transcriptional changes observed at D21 prior to any significant changes in microCT parameters, suggest that OAA, at least in part, effects changes during the early stages of osteogenic differentiation. We hope we have made these considerations more clear in our discussion, while at the same time acknowledging the limitations of the studies presented.